# Physiological and biochemical characterization of trypsin from *Neocaridina denticulata sinensis* and its roles in ontogenesis and immune response

Dandan Feng[1], Yakun Song[1], Zuqi Wu[1], Wuruo Liu[1], Yuting Pu[1,2], Yangcan Gao[1], Yuying Sun[1,2,3]*, Jiquan Zhang[1]*

1 School of Life Sciences/Hebei Basic Science Center for Biotic Interaction, Hebei University, Baoding, China, 2 Key Laboratory of Microbial Diversity Research and Application of Hebei Province, Baoding, China, 3 Engineering Research Center of Microbial Breeding and Conservation, Baoding, China

* sunyuying125@hbu.edu.cn (YS); zhangjiquan@hbu.edu.cn (JZ)

## Abstract

Trypsin, a canonical serine protease in crustaceans, plays a crucial role in ontogeny and antibacterial defense. Whether these biological functions correlate with its catalytic characteristics remains unresolved in the freshwater shrimp *Neocaridina denticulata sinensis*. To address this gap, we characterized a trypsin gene from *N. denticulata sinensis* (*NdTryp*) and assessed both its biological roles and its prospective utility. *NdTryp* was predominantly expressed in the hepatopancreas, where it localized to storage cells (R-cells) and tubule-lining epithelial cells (ECTs). Across development, *NdTryp* transcripts were essentially absent during early embryogenesis but rose sharply at late stages, temporally coincident with hatching and the onset of feeding. After a challenge with *Vibrio parahaemolyticus*, the expression of *NdTryp* was induced, with the expression level significantly increased relative to the baseline expression level. RNA interference-mediated knockdown rendered shrimp more susceptible to infection and was accompanied by extensive hepatopancreatic injury, including epithelial detachment and disruption of the basement membrane. Biochemically, recombinant NdTryp (rNdTryp) displayed proteolytic activity over a broad temperature and pH span. Activity was differentially tuned by metal ions, with several divalent cations producing marked enhancement, whereas ferric iron exerted strong inhibition. Overall, our results showed that NdTryp functions as a multifunctional protease involved in both late embryonic development and innate antimicrobial defense. Furthermore, the robust stability of rNdTryp underscores its potential as an aquafeed additive and candidate for enzymatic biotransformation.

**Data availability statement:** All relevant data are within the paper and its Supporting Information file.

**Funding:** This work was supported by the National Natural Science Foundation of China (Grant Nos. 32172954, 32373121) and the Natural Science Foundation of Hebei Province of China (D2023201002). The funders (Jiquan Zhang) had a role in study design, data collection and analysis, decision to publish, or preparation of the manuscript.

**Competing interests:** The authors have declared that no competing interests exist.

## 1. Introduction

Proteases are among the most abundant and diverse enzymes in biological systems. In addition to catalyzing protein degradation, they play essential regulatory roles by selectively modulating specific functional proteins to sustain cellular homeostasis [1]. In crustaceans, the tissues in which digestion occurs are mainly the hepatopancreas and intestine. This is because these two tissues contain large amounts of protein-hydrolyzing enzymes [2]. Proteolytic activity is primarily mediated by serine proteases, among which trypsin (EC 3.4.21.4) constitutes the major digestive endopeptidase [2,3]. Trypsin hydrolyzes proteins, resulting in damage to tissues. Therefore, trypsin is produced in an inactive precursor form — trypsinogen [4]. Mature trypsin can hydrolyze proteins at arginine and lysine in the substrate protein sequence [5,6]. Trypsin and trypsin-like enzymes also play important physiological roles, such as blood clotting, fibrinolysis, fertilization, embryonic development, and immune response [7–9]. Furthermore, trypsin is an antimicrobial peptide prodefensin convertase and also plays an important role in the regulation of innate immunity [10].

In the global aquaculture industry, shrimp represents a commodity of substantial economic importance. On the one hand, as a common pathogen of crustaceans, *Vibrio parahaemolyticus* can result in acute hepatopancreatic necrosis (AHPND) syndrome or early mortality syndrome, leading to severe economic losses in the shrimp industry in China [11]. In addition, human consumption of food infected with *V. parahaemolyticus* may lead to foodborne gastroenteritis [12,13]. However, *V. parahaemolyticus* has emerged and evolved antibiotic resistance over the past few decades due to the overuse of antibiotics in humans and aquaculture systems [14]. Therefore, research into new biological methods to prevent and control the invasion of *V. parahaemolyticus* is necessary.

On the other hand, the post-harvest shelf life of economic crustaceans remains severely constrained by texture deterioration. The primary cause of this process is trypsin-catalyzed degradation of muscle tissue, which severely limits the shelf life of shrimp during refrigerated transportation [15–17]. Although the presence of trypsin has adverse effects on the refrigerated transportation of shrimp [18], it holds application potential as a prospective biocatalyst. Therefore, an understanding of the physicochemical properties of trypsin is essential for the meat storage of economically important crustaceans. *Neocaridina denticulata sinensis* belongs to the Decapoda and is a small freshwater economic shrimp. It is used as a model animal in physiology and ecology due to its advantageous traits, including a high reproductive rate, short life cycle, and simple husbandry requirements [19–21].

Trypsin has been characterized in multiple crustaceans (e.g., *Macrobrachium carcinus*, *Panulirus argus*, and *Litopenaeus vannamei*), revealing species-specific isoforms and functional diversity in digestion and immune-related processes [2,9,22]. However, fundamental information on trypsin in *N. denticulata sinensis* (NdTryp) remains largely unknown. Critical gaps include the full-length gene sequence and structural features, tissue and cellular expression patterns, developmental expression dynamics, functional involvement in antibacterial defense, and biochemical properties, including metal-ion sensitivity. Therefore, *N. denticulata sinensis* was selected as

a model to study the function of trypsin in decapod crustaceans in this study. The full-length cDNA sequence of the trypsin (*NdTryp*) gene was obtained from the *N. denticulata sinensis*, and the expression profiles were studied in different tissues, embryos at different developmental stages, and post-*V. parahaemolyticus* challenge. *In situ* hybridization (ISH) experiments determined that *NdTryp* mRNA was expressed mainly in storage cells (R-cells) and epithelial cells lining the tubules (ECTs) of hepatopancreatic tissue. To achieve the NdTryp for industrial applications, recombinant NdTryp (rNdTryp) was prepared using a prokaryotic expression system. Then, the enzymatic properties of the purified product were studied for pH, temperature, and metal ion stability. These findings provide insights into the roles of NdTryp in embryonic development, immune defense, and potential industrial applications.

## 2. Materials and methods

### 2.1. Experimental animals

Healthy shrimp (*N. denticulata sinensis*, body length, $2.0 \pm 0.5$ cm) were obtained from a local market in Anxin County, Baoding, Hebei Province, China. Before experimentation, animals were acclimated for 7 days in recirculating aquaria maintained at $25 \pm 1$ °C under a 12: 12 h light: dark cycle, and were fed a commercial diet (Sera Shrimp Natural; Kunshan) twice daily. All procedures were reviewed and approved by the Institutional Animal Care and Use Committee of Hebei University and were conducted in accordance with the ARRIVE guidelines (https://arriveguidelines.org). As the study involved a non-endangered invertebrate species, formal ethics approval was not required under local legislation (China) or institutional policy.

### 2.2. Gene cloning

The trypsin gene sequence (NdTryp) was identified from a previously published full-length transcriptome [20], and gene-specific primers (S1 File) were designed using Primer Premier 5.0. The full-length cDNA of *NdTryp* was amplified via PCR using Ex Taq˚ DNA Polymerase (TaKaRa, Dalian, China). The resulting PCR product was then ligated into the pMD19-T vector (TaKaRa, Dalian, China) and subsequently transformed into DH5α competent cells (Sangon Biotech, Beijing, China). Individual colonies were selected for sequencing at GENEWIZ Co., Ltd. (Tianjin, China). The sequenced open reading frame (ORF) was mapped against genomic data (unpublished) to determine exon-intron boundaries.

### 2.3. Bioinformatics analysis of NdTryp

The analysis of the full-length cDNA sequence of *NdTryp* was based on the sequencing results and was performed using the NCBI BLAST program (https://blast.ncbi.nlm.nih.gov/Blast.cgi, accessed April 22, 2024). Predicting speculated functional sites within NdTryp using the Prosite database (https://prosite.expasy.org/, accessed April 22, 2024). Determine theoretical isoelectric point (pI) using ExPASy online tool (https://web.expasy.org/protparam/, accessed April 22, 2024). Other protein feature analyses were conducted using SMART online tools, including abnormal homologs, homologs with known structures, PFAM domain identification, signal peptide prediction, and internal duplication identification (http://smart.embl.de/smart/set_mode.cgi?NORMAL=1, accessed April 24, 2024). Conserved sequences were identified through multiple sequence alignment using BioEdit software. Using insects as outgroups, a phylogenetic tree was constructed using the Neighbor-joining (NJ) method in MEGA 7.0 software [23]. Three-dimensional structure modeling of the NdTryp protein using the SWISS-MODEL online server (https://swissmodel.expasy.org/interactive, accessed April 25, 2024), and visualized the obtained model using PyMOL software.

### 2.4. Methodology of RNA interference (RNAi)

Double-stranded RNA (dsRNA) was synthesized according to our previously described method as our previously described [24]. Briefly, gene-specific primers targeting the ORF of *NdTryp* and *EGFP* were designed, each adding T7

promoter sequence at the 5′ end (S1 File). The ds*NdTryp* and ds*EGFP* were subsequently synthesized and purified using the TranscriptAid T7 High Yield Transcription Kit (Thermo Scientific, Waltham, MA, USA) according to the manufacturer's instructions. Shrimp received dsNdTryp injections (1,000 ng/µL, 5 µL per individual) into the second abdominal segment. An equivalent dose of ds*EGFP* was injected into a separate group of shrimp under identical conditions to serve as a negative control.

## 2.5. Experimental challenge with V. parahaemolyticus

*V. parahaemolyticus* strain was used in this study, containing virulence genes *pirA*[VP] and *piraB*[VP], which can cause AHPND in shrimp. In this study, the *V. parahaemolyticus* challenge test used previously described methods with modifications [25]. In brief, *V. parahaemolyticus* was inoculated onto a 25‰ salinity of high-salt LB liquid medium and incubated overnight at 37 °C and a shaking speed of 200 r/min. To obtain logarithmic-phase *V. parahaemolyticus* cultures, bacteria were subcultured at 1% (v/v) for secondary amplification. The bacteria were collected by centrifugation at a speed of 2,000 r/min for 5 min, and then resuspended in 1 x PBS buffer (Solarbio, Beijing, China). The concentration was adjusted to $1 \times 10^6$ CFU/mL with 1 x PBS buffer. Ninety shrimp were evenly divided into three groups, and each shrimp in each group was injected with 5 µL of live *V. parahaemolyticus*. Three individuals per group were randomly sampled at 0, 6, 12, 24, 48, and 72 h post-challenge, representing three biological replicates for each time point. To investigate the expression profile changes of *NdTryp* after *V. parahaemolyticus* infection, the hepatopancreas was selected as a typical tissue for the study, considering its immune tissue characteristics. To verify the immune function of NdTryp, an equal amount of *V. parahaemolyticus* was injected into the shrimp after 24 h of knockdown of *NdTryp* expression for histological structure research. *V. parahaemolyticus* stimulation experiment randomly selected 3 individuals at each time point as biological replicates ($n = 3$), and RNA was independently extracted from the hepatopancreases of each individual and reverse transcribed separately.

## 2.6. Quantitative real-time PCR (qPCR) assay

To determine tissue-specific *NdTryp* expression, total RNA was extracted from the intestines, muscles, ovary, gills, epidermis, testis, stomachs, hearts, eyestalks, hepatopancreases, and abdominal nerves of three shrimp. To investigate the role of *NdTryp* in embryonic development, shrimp embryos were collected at various developmental stages, including blastocyst stage, cleavage stage, gastrula stage, anterior nauplius stage, posterior nauplius stage, prophase of facetted eye pigmentation, anaphase of facetted eye pigmentation, zoea stage, and larvae stage. Each sample was pooled from three independent individuals ($n = 3$ biological replicates) and immediately frozen in liquid nitrogen for RNA extraction. According to the operation manual provided by the manufacturer, HiScript® III 1st Strand cDNA Synthesis Kit (+gDNA wiper) (Vazyme, Nanjing, China) was used to reverse transcribe RNA into cDNA. *18S* ribosomal RNA (GenBank accession number: OP185352) served as an internal control. All primers used for qPCR are listed in the S1 File. The qPCR assays were conducted for *NdTryp* and *18S* ribosomal RNA using a Bioer LineGene 9600 system (Hangzhou, China). The amplification protocol consisted of an initial denaturation step at 95 °C for 3 min, followed by 40 cycles of denaturation at 95 °C for 10 s, annealing at 60 °C for 10 s, and extension at 72 °C for 15 s. Amplification specificity was verified via melting-curve analysis, which was performed from 60 °C to 95 °C at a ramp rate of 0.4 °C/s. To monitor potential contamination and confirm assay specificity, no-template controls (NTCs) were routinely included in each qPCR run. For RNAi experiments, knockdown efficiency detection was performed on 3 independent individuals ($n = 3$, biological replicates) at each time point, and 3 technical replicates were performed for qPCR of each sample. Then, each cDNA sample was tested in qPCR with 3 wells (technical replicates = 3), and the final result was expressed as the mean ± SD of the 3 biological replicates. Relative quantification of *NdTryp* expression levels was determined using the comparative CT method [26].

## 2.7. Methodology of hematoxylin-eosin (H&E) staining

To characterize the morphology of the *N. denticulata sinensis* hepatopancreas, H&E staining was performed. Hepatopancreases were rapidly excised using sterile forceps and fixed in 4% paraformaldehyde solution (BOSTER, Wuhan, China) for 24 h. Following fixation, tissues were dehydrated through a graded ethanol (30%, 50%, 75%, 85%, 95%, and 100%) at 4 °C, with each step lasting 1 h. The tissues were immersed in a 1:1 (v/v) mixture of xylene and paraffin wax at 65 °C for 30 min, followed by immersion in pure paraffin wax at 70 °C for 1.5 h. Then, the samples were embedded in paraffin, sectioned at a thickness of 5 μm, and mounted onto glass slides (CITOTEST, Nanjing, China). Finally, sections were stained with H&E (Biotopped, Beijing, China) and examined under an inverted research microscope (Nikon, Japan).

## 2.8. In situ hybridization (ISH) of NdTryp mRNA-expressing cells

To investigate the cellular localization of *NdTryp* mRNA in hepatopancreases, ISH was used for analysis in this study. The ISH experiment of hepatopancreas tissue slices was conducted according to previous literature reports with modifications [25]. In brief, hepatopancreatic tissues were dissected using sterile tweezers and immediately fixed in 4% paraformaldehyde (BOSTER, Wuhan, China) for 24 h at 4 °C. After fixation, the samples were dehydrated using a gradient of methanol (MACKLIN, Shanghai, China) series and embedded in paraffin. Then, the embedded tissue blocks were prepared into 5 μm-thick serial sections using a microtome and mounted onto glass slides. The slices were first subjected to permeabilization treatment with 2 μg/mL proteinase K (Aladdin, Shanghai, China) at 37 °C for 30 min, followed by pre-hybridization at 55 °C for 4 h. The slices were incubated and hybridized with 50 ng/μL of digoxin (DIG) labeled *NdTryp* mRNA-specific RNA probe (Roche Diagnostics, Mannheim, Germany) at 55 °C for 16 h. After hybridization, the slices were co-incubated with anti-digoxin alkaline phosphatase (AP) labeled Fab fragments (Roche Diagnostics, Mannheim, Germany) at 4 °C for 14 h. Finally, the BCIP/NBT colorimetric assay kit (Solarbio, Beijing, China) was used for colorimetric detection and observation of hybridization signals.

## 2.9. Prokaryotic expression, purification, and enzymatic activity assay of recombinant NdTryp (rNdTryp)

Based on our previously constructed vector pCT7-CHISP6H of protein expression [27], a pair of primers pCT7-CHISP6H-NdTryp-F/R (S1 File) was designed for PCR amplification, and sequencing plasmid pMD19-NdTryp was used as the template. The PCR products were recovered and ligated into pCT7-CHISP6H using ABclonal MultiF Seamless Assembly Mix (ABclonal® Technology, Wuhan, China) at 16 °C for 1 h. The ligated product (pCT7-CHISP6H-NdTryp) was transformed into *Escherichia coli* DH 5α competent cells for DNA sequencing. The plasmid of the positive clone was transferred into the BL21-Star (DE3) pLysS competent cells (WEIDI, Shanghai, China) to express proteins, and the protein purification was carried out as previously described [28]. Briefly, a seed culture was grown for 12 h, after which 1% v/v was inoculated into 200 mL of LB medium supplemented with ampicillin (100 μg/mL) and chloramphenicol (100 μg/mL). When the $OD_{600}$ reached 0.6–0.8, protein expression was induced by adding isopropyl β-D-1-thiogalactopyranoside (IPTG) (MACKLIN, Shanghai, China), followed by incubation at 37 °C for 6 h. The bacterial cells were harvested by centrifugation at a speed of 7,000 r/min for 30 min at 4 °C. The cell pellet was resuspended in PBS buffer (0.15 M NaCl, 10% (w/v) glycerol, pH 7.4) and lysed using an ultrasonic homogenizer (SCIENTZ, Ningbo, China). The supernatant was collected by centrifugation at a speed of 12,000 r/min for 60 min at 4 °C, and the recombinant protein was purified using a $Ni^{2+}$-NTA column (Cytiva, Danaher) affinity chromatography. The enzymatic activity was determined using the trypsin assay kit (A080-2–2) from Nanjing Jian Cheng Bioengineering Institute. According to the manufacturer's protocol, under the conditions of pH 8.0 and temperature 37 °C, trypsin causes a change in absorbance of 0.003 per milligram of protein per minute, defined as one unit (U) of enzyme activity.

## 2.10. Biochemical characterization of rNdTryp

To delineate the temperature dependence, the enzyme solution (200 μL, 1 mg/mL) was incubated for 1 h at 30–60 °C, with three replicates at each temperature. Relative activity was subsequently read out using a commercial trypsin activity

assay kit. For pH profiling, equal aliquots of enzyme solution (200 μL, 1 mg/mL) were incubated for 1 h at 35 °C in buffers spanning pH 3.0–11.0. Buffer systems were matched to the intended pH range — citrate (pH 3.0, 4.0, 5.0), sodium phosphate (pH 6.0, 7.0, 8.0), Tris-HCl (pH 9.0), and sodium carbonate–sodium hydroxide (pH 10.0, 11.0). In parallel, metal-ion effects were evaluated by mixing the enzyme solution with $K^+$, $Ca^{2+}$, $Mg^{2+}$, $Cd^{2+}$, $Fe^{3+}$, $Li^+$, $Cu^{2+}$, $Co^{2+}$, $Ba^{2+}$, or $Mn^{2+}$ to a final ion concentration of 20 mM and incubating the mixtures at 37 °C for 1 h. All activity measurements were obtained in triplicate (technical replicates) to limit assay-level variation. Relative enzymatic activity was calculated using the same trypsin assay kit throughout.

### 2.11. Statistical analysis

For experiments involving animals, $n$ denotes the number of individual shrimp (biological replicates), with each animal treated as an independent biological unit. Data were reported as mean±SD. Statistical analyses were conducted in SPSS (version 19.0, USA). Comparisons across multiple groups used one-way ANOVA followed by Duncan's multiple-range test, whereas prespecified pairwise contrasts were evaluated with two-tailed $t$ tests. Data handling and figure preparation were performed in GraphPad Prism (version 9.0.2; San Diego, CA, USA).

## 3. Results

### 3.1. Characteristics of the NdTryp gene and its deduced amino acid (aa) in N. denticulata sinensis

The full-length cDNA sequence of *NdTryp* was obtained through Sanger sequencing of *N. denticulata sinensis* and deposited in GenBank under accession number **PP944486**. Compared with the genomic data, the target gene contains three exons and two introns (S2 File and S3 File). The sequences at the exon-intron boundaries conformed to the typical eukaryotic splice sites, including an invariant GT at the intron 5′ boundary and an invariant AG at its 3′ boundary [29]. As shown in Fig 1, the complete nucleotide sequence of *NdTryp* was 906 bp, including 33 bp 5′-UTR, 801 bp ORF (S4 File), and 72 bp 3′-UTR. The ORF encoded a 266-amino acid protein with a predicted pI of 4.26. The deduced NdTryp protein contained a 233-residue Tryp_SPc domain predicted by SMART. Site predictions revealed the presence of three active sites (His 74, Asp 125, and Ser 218) (Fig 2). A three-dimensional model was constructed using an X-ray template of crayfish hepatopancreas trypsin (PDB ID: 4bnr.1.A) and was assessed by the QMEAN scoring function. The GMQE and QMEANDisCo Global scores were 0.84 and 0.89±0.05, respectively, suggesting that the modeling was accurate. The result indicated that the NdTryp protein was a monomer and contained $Ca^{2+}$ binding sites (Asp 185, Asp 199, Met 201, and Glu 253). A three-dimensional image of NdTryp shows the two β-barrels making up the fold, connecting α-helices, surface loops, and the catalytic residues (His 74, Asp 125, and Ser 218) (Fig 3). The phylogenetic analysis indicated that NdTryp is embedded in the branch of Malacostraca belonging to trypsin (Fig 4).

### 3.2. qPCR analysis of the distribution of NdTryp in different tissues and during shrimp ontogenesis

The melt curve analysis revealed a single peak for the *NdTryp* gene, indicating the qPCR product has a high degree of specific amplification. The primer NdTryp-qPCR-F/R amplification sequence covered the conserved domain of NdTryp, with a length of 121 bp. To verify the specificity and accuracy of the primers, their amplification efficiency was validated (S5 File and S6 File). The calculated amplification efficiency (E) was 96.65% ($R^2$=0.9971), demonstrating that the NdTryp-qPCR-F/R primer exhibited high efficiency and accuracy across various concentrations. The expression patterns of the *NdTryp* in various tissues showed that *NdTryp* was detected in all tested tissues (Fig 5a), which demonstrated *NdTryp* mRNA to be constitutively expressed in *N. denticulata sinensis*. The results presented a particularly high expression in hepatopancreas (76533.5-fold), contrasting with findings in *Pinctada fucata martensii* [30]. Testis exhibited the second-highest expression. Subsequently, comparable expression levels were observed in the ovary, abdominal nerve, and stomachs, without significant inter-tissue variation ($p > 0.05$). The lowest expression was detected in the intestine,

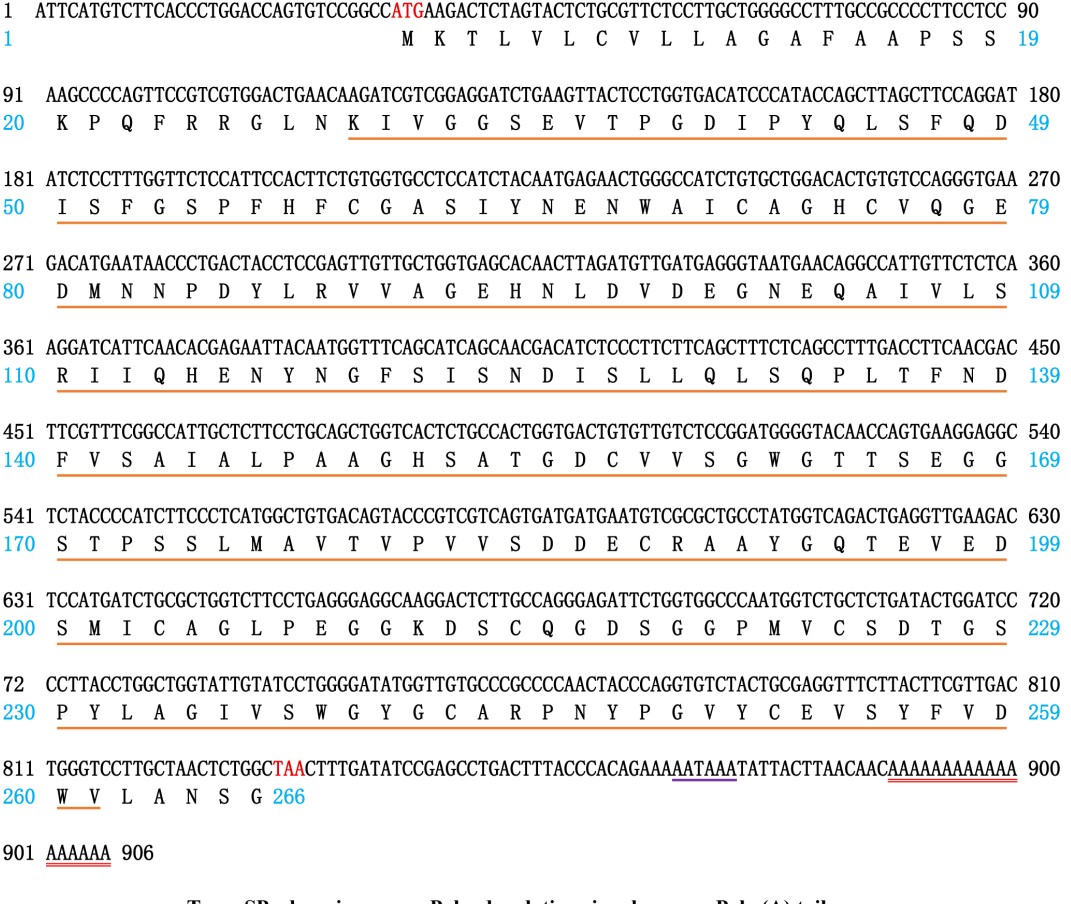

```
1   ATTCATGTCTTCACCCTGGACCAGTGTCCGGCCATGAAGACTCTAGTACTCTGCGTTCTCCTTGCTGGGGCCTTTGCCGCCCCTTCCTCC  90
1                                         M   K   T   L   V   L   C   V   L   L   A   G   A   F   A   A   P   S   S   19

91  AAGCCCCAGTTCCGTCGTGGACTGAACAAGATCGTCGGAGGATCTGAAGTTACTCCTGGTGACATCCCATACCAGCTTAGCTTCCAGGAT  180
20   K   P   Q   F   R   R   G   L   N   K   I   V   G   G   S   E   V   T   P   G   D   I   P   Y   Q   L   S   F   Q   D   49

181 ATCTCCTTTGGTTCTCCATTCCACTTCTGTGGTGCCTCCATCTACAATGAGAACTGGGCCATCTGTGCTGGACACTGTGTCCAGGGTGAA  270
50   I   S   F   G   S   P   F   H   F   C   G   A   S   I   Y   N   E   N   W   A   I   C   A   G   H   C   V   Q   G   E   79

271 GACATGAATAACCCTGACTACCTCCGAGTTGTTGCTGGTGAGCACAACTTAGATGTTGATGAGGGTAATGAACAGGCCATTGTTCTCTCA  360
80   D   M   N   N   P   D   Y   L   R   V   V   A   G   E   H   N   L   D   V   D   E   G   N   E   Q   A   I   V   L   S   109

361 AGGATCATTCAACACGAGAATTACAATGGTTTCAGCATCAGCAACGACATCTCCCTTCTTCAGCTTTCTCAGCCTTTGACCTTCAACGAC  450
110  R   I   I   Q   H   E   N   Y   N   G   F   S   I   S   N   D   I   S   L   L   Q   L   S   Q   P   L   T   F   N   D   139

451 TTCGTTTCGGCCATTGCTCTTCCTGCAGCTGGTCACTCTGCCACTGGTGACTGTGTTGTCTCCGGATGGGGTACAACCAGTGAAGGAGGC  540
140  F   V   S   A   I   A   L   P   A   A   G   H   S   A   T   G   D   C   V   V   S   G   W   G   T   T   S   E   G   G   169

541 TCTACCCCATCTTCCCTCATGGCTGTGACAGTACCCGTCGTCAGTGATGATGAATGTCGCGCTGCCTATGGTCAGACTGAGGTTGAAGAC  630
170  S   T   P   S   S   L   M   A   V   T   V   P   V   V   S   D   D   E   C   R   A   A   Y   G   Q   T   E   V   E   D   199

631 TCCATGATCTGCGCTGGTCTTCCTGAGGGAGGCAAGGACTCTTGCCAGGGAGATTCTGGTGGCCCAATGGTCTGCTCTGATACTGGATCC  720
200  S   M   I   C   A   G   L   P   E   G   G   K   D   S   C   Q   G   D   S   G   G   P   M   V   C   S   D   T   G   S   229

72  CCTTACCTGGCTGGTATTGTATCCTGGGGATATGGTTGTGCCCGCCCCAACTACCCAGGTGTCTACTGCGAGGTTTCTTACTTCGTTGAC  810
230  P   Y   L   A   G   I   V   S   W   G   Y   G   C   A   R   P   N   Y   P   G   V   Y   C   E   V   S   Y   F   V   D   259

811 TGGGTCCTTGCTAACTCTGGCTAACTTTGATATCCGAGCCTGACTTTACCCACAGAAAAATAAATATTACTTAACAACAAAAAAAAAAAA  900
260  W   V   L   A   N   S   G   266

901 AAAAAA  906
```

|  |  |  |
|---|---|---|
| **Tryp_SPc domain** | **Polyadenylation signal** | **Poly (A) tails** |

**Fig 1. Schematic diagram representing the full-length cDNA and deduced amino acid sequence of *NdTryp*.** The red letters indicate the start site (ATG) and stop codon (TAA). In the aa sequence, the Tryp_SPc domain and polyadenylation signal are marked with different symbols. The poly(A) tail is indicated by a red double underline at the end of the sequence.

muscle, epidermis, gill, heart, and eyestalk, which also showed no significant inter-tissue variation ($p > 0.05$) (S7 File). *NdTryp* expression was not detected uniformly throughout the nine distinct embryonic developmental stages in shrimp. *NdTryp* expression was undetectable during early embryonic stages (blastocyst stage, cleavage stage, gastrula stage, anterior nauplius stage, and posterior nauplius stage), but emerged at the early eye-pigmentation stage. Upon reaching the late eye-pigmentation stage, the expression of *NdTryp* increased significantly and remained at high expression levels until the larval stage (Fig 5b). There was no significant difference ($p > 0.05$) in the expression of *NdTryp* at anaphase of facetted eye pigmentation, zoea stage, and larvae stages, but the expression levels at these stages were approximately 390,000-fold higher than that of prophase of facetted eye pigmentation.

### 3.3. Response of NdTryp mRNA expression under V. parahaemolyticus challenge

The expression profiles of *NdTryp* showed a change from 0 to 72 h after *V. parahaemolyticus* challenge (Fig 6a). *NdTryp* expression was progressively upregulated through 48 h post-challenge, relative to baseline (0 h). Peak expression occurred at 24 h post-challenge and persisted through 36 h, approximately three times higher than at 0 h. Expression exhibited cyclical fluctuations between 36 and 72 h, returning to baseline levels by 72 h ($p > 0.05$ vs. 0 h). To elucidate

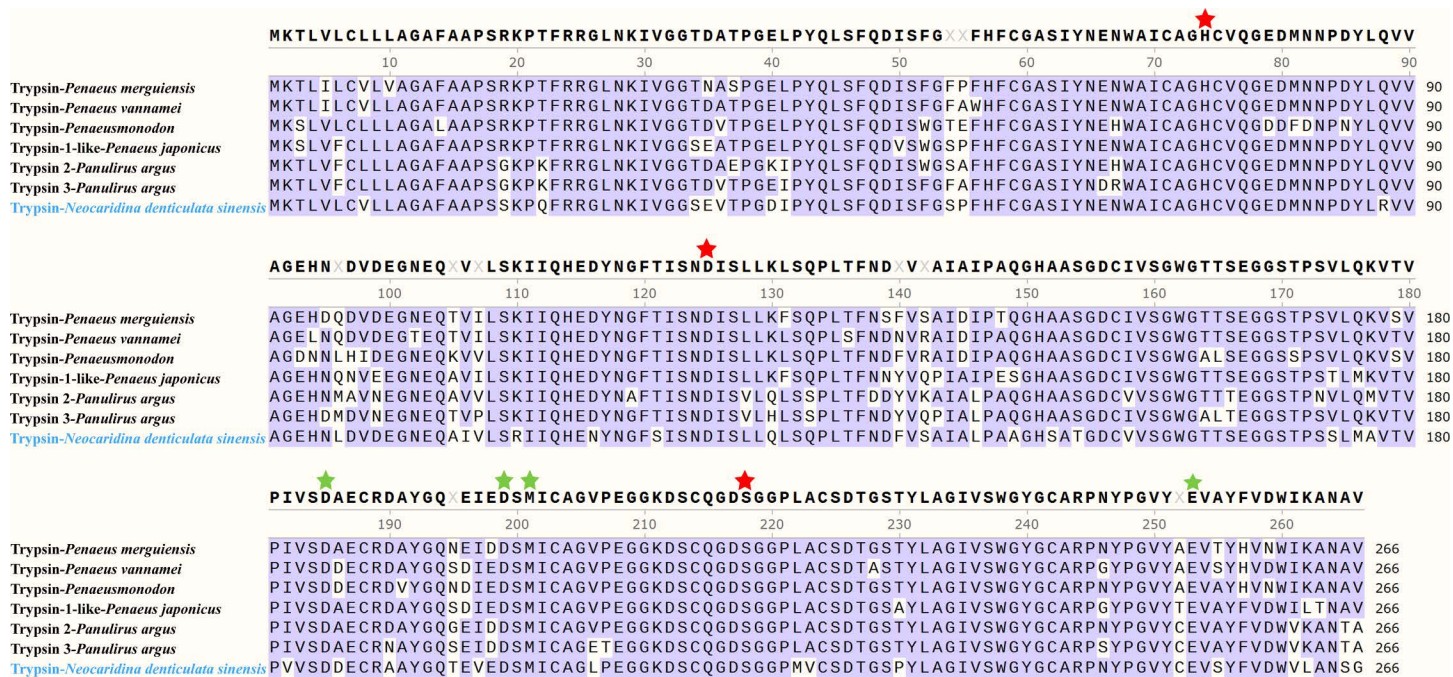

**Fig 2. Alignment of the amino acid sequence of NdTryp with the known trypsin or trypsin-like.** The red and green pentagrams represent conserved catalytic and calcium ion binding sites, respectively.

NdTryp immune function, expression was suppressed via RNAi. Before the functional challenge experiment, a preliminary experiment was conducted to determine the optimal time point for RNAi efficiency after dsRNA injection. The results indicated that the expression of *NdTryp* was most significantly suppressed at 24 h post-injection (Fig 6b). In the control group, the structure of hepatopancreatic cells remained intact (Fig 6c and S8 File). However, after 72 h of *V. parahaemolyticus* challenge following knockdown of *NdTryp*, partial necrosis was observed in the hepatopancreatic tissue, with necrotic debris accumulating in tubule lumens. Detachment of the ECT, secretory cells (B-cell), and R-cell from the basement membrane (BM) was evident, leading to a loss of discernible cellular architecture. Furthermore, localized ruptures of the BM resulted in cellular disintegration (Fig 6d and S9 File).

### 3.4. Localization of NdTryp mRNA expression in hepatopancreas

ISH detected the cellular expression of *NdTryp* mRNA in the hepatopancreas. Sense-probe controls showed no signal, whereas antisense probes yielded strong positive signals for *NdTryp*-expressed cells (Fig 7a and S10 File), whereas the experimental group showed strongly positive signals for *NdTryp*-expressed cells (Fig 7b and S11 File). As shown in Fig 7b, the expression of *NdTryp* mRNA varied in different areas of the hepatopancreas. The positive signals were primarily detected in hepatopancreatic cells of two types of shrimp, namely R-cells and ECTs. Conversely, the positive signals were not observed in B-cells. Therefore, the strength of positive signals was also different in a hepatopancreas cell, suggesting that *NdTryp* might act in various roles in exerting biological functions.

### 3.5. Purification of rNdTryp and detection of enzymatic activity

SDS-PAGE analysis revealed that the purified rNdTryp migrated as a single band with an apparent molecular weight of approximately 35 kDa (Fig 8 and S12 File). The molecular weight of rNdTryp was consistent with that of trypsin

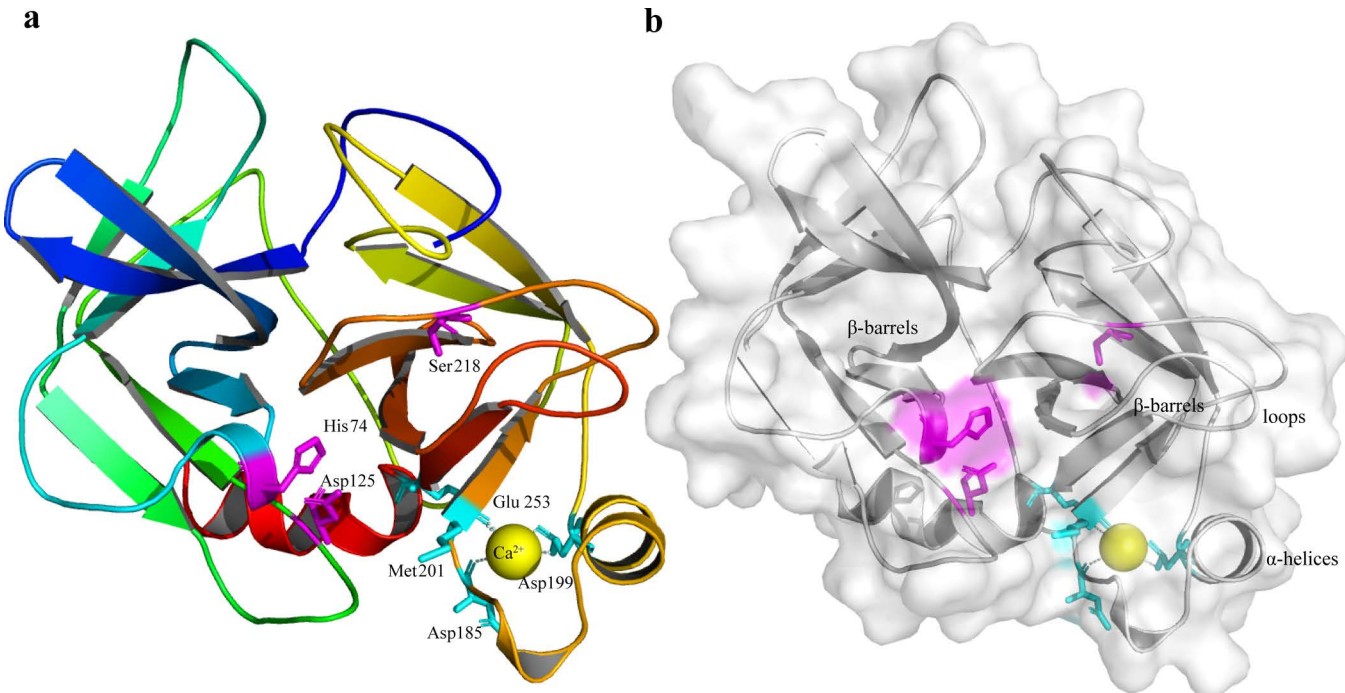

**Fig 3. The predicted three-dimensional protein model for a NdTryp based on crayfish hepatopancreas trypsin (PDB ID: 4bnr.1. A) by the SWISS-MODEL server. (a)** The three-dimensional physical structure of the NdTryp protein is presented as a monomer. The catalytic residues (His 74, Asp 125, and Ser 218) are represented in purple. A yellow sphere represents a calcium ion that binds to four amino acid residues (Asp 185, Asp 199, Met 201, and Glu 253, respectively) shown in light blue. **(b)** The three-dimensional physical structure of NdTryp shows two β-barrels and an α-helix.

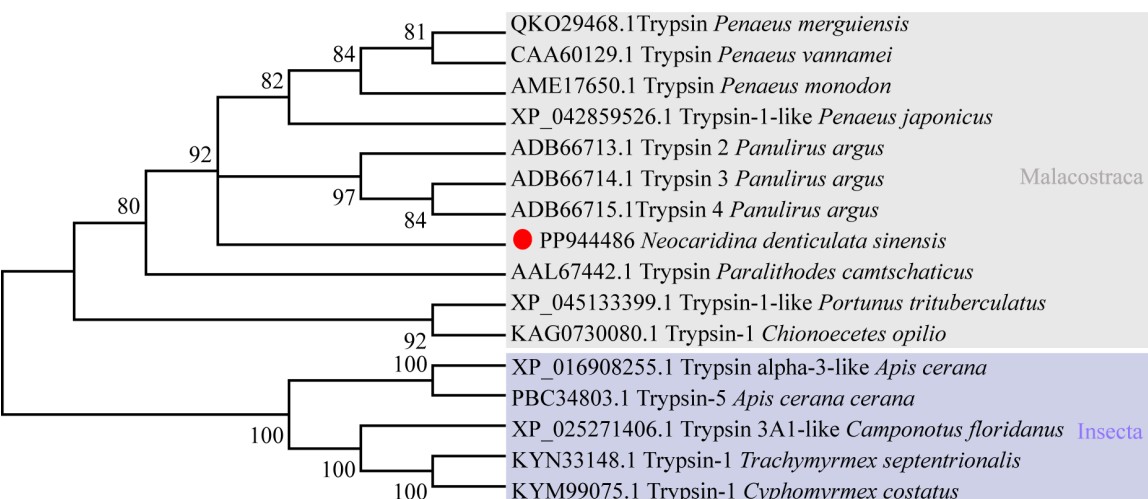

**Fig 4. Phylogenetic analysis of trypsin protein among various species using the neighbor-joining method with a bootstrap value of 1,000, and Insecta is selected as one outgroup.** The NdTryp was displayed with a red circle.

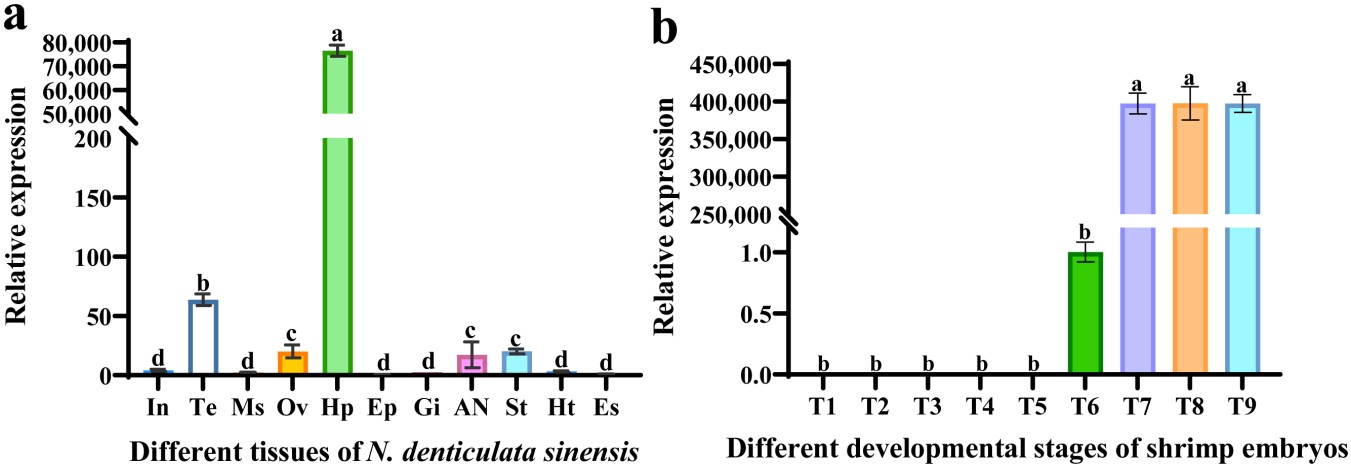

**Fig 5. *NdTryp* mRNA expression in different tissues and during ontogenesis. (a)** Tissue distribution of *NdTryp* mRNA. The tested tissues include abdominal nerve (AN), muscles (Ms), intestines (In), hepatopancreases (Hp), testis (Te), gills (Gi), hearts (Ht), stomachs (St), ovary (Ov), eyestalk (Es), and epidermis (Ep). In these cases, the different letters (a, b, c, d) on the line designate statistical significance ($p < 0.05$), and values sharing a common letter are not significantly different ($p > 0.05$). Histological section of shrimp hepatopancreas and *in situ* hybridization of *NdTryp* mRNA expression in the hepatopancreas. **(b)** Expression of *NdTryp* in shrimp embryos at nine different developmental stages: Blastocyst stage (T1), cleavage stage (T2), gastrula stage (T3), anterior nauplius stage (T4), posterior nauplius stage (T5), prophase of facetted eye pigmentation (T6), anaphase of facetted eye pigmentation (T7), zoea stage (T8) and larvae (T9).

previously identified in *Thenus orientalis* [31]. The enzymatic activity was determined to be $233.78 \pm 4.07$ U/mL using the commercial kit.

### 3.6. Enzymatic characterizations of rNdTryp

The optimal pH for rNdTryp activity was approximately 8.0, differing slightly from the optimum of 7.0 reported for trypsin in *P. argus* [9]. The enzyme retained high activity across a broad alkaline pH range but was rapidly inactivated under moderately acidic conditions (Fig 9a). The optimal temperature for rNdTryp was 35 °C, similar to that of trypsin in *Coryphaenoides pectoralis* [32]. The enzyme activity is significantly affected at 60 °C, with a relative enzyme activity of approximately 7.18% (Fig 9b). In the presence of 20 mM $Cd^{2+}$, $Cu^{2+}$, and $Mn^{2+}$, rNdTryp exhibited the highest enzyme activity. $K^+$, $Mg^{2+}$, and $Fe^{3+}$ inhibited rNdTryp relative enzyme activity, with $Fe^{3+}$ inhibiting relative enzyme activity by about 71% (Table 1).

## 4. Discussion

### 4.1. Structural features and evolutionary conservation of NdTryp

Trypsin has a catalytic triad consisting of residues (His 57, Asp 102, and Ser 195), which cleaves the peptide bonds on the carboxyl groups of Arg and Lys residues at the position of the protein substrate [9]. In contrast to the previous report [9], the catalytic triad residues of NdTryp are His 74, Asp 125, and Ser 218. The amino acid categories of the trypsin active site are conserved, but the exact position is slightly variable. The calcium ion binding sites contain two aspartic acid residues, presumably because aspartic acid is the negative charge at physiological pH. According to the principle of mutual attraction between positive and negative charges, the negative charge of aspartic acid residues binds better with the positive charge of calcium ions. In *Tribolodon hakonensis* [33], the structural properties of the N-terminal region and calcium ion-binding region of trypsin are closely related to thermostability. Two six-stranded β-barrel structural domains are stacked together, and catalytic residues (His 74, Asp 125, Ser 218) are located at the junction of the two β-barrels. There is a report that the β-barrel structure facilitates the maintenance of the thermal stability of trypsin [34].

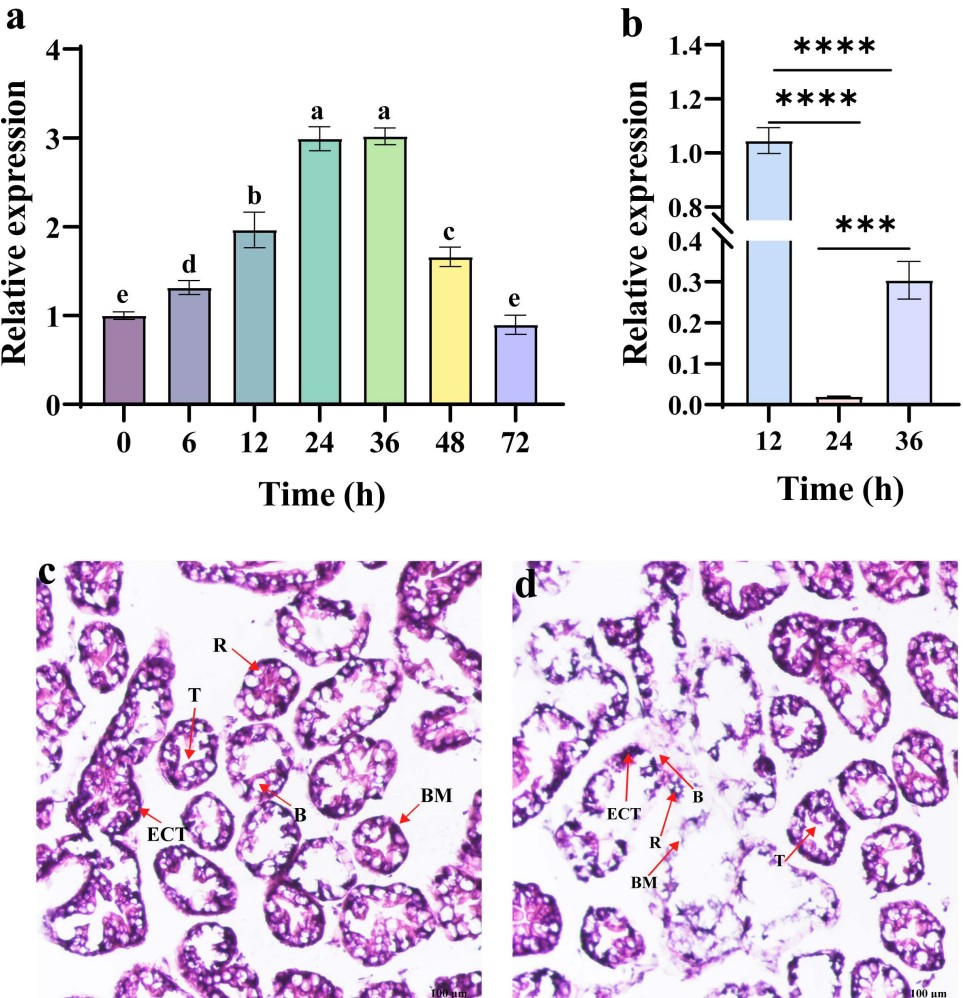

**Fig 6. Expression profile of *NdTryp* after *V. parahaemolyticus* stimulation and histomorphology of hepatopancreas challenged by *V. parahaemolyticus* after *NdTryp* knockdown.** (a) Expression profiles of *NdTryp* in the hepatopancreas of *N. denticulata sinensis* at different times after the shrimp were challenged with *V. parahaemolyticus*. The data were the means of three independent experiments ± S.E., and statistically significant differences between the mean values were detected with one-way ANOVA. The different letters (a, b, c, d, and e) on the columns designate statistical significance ($p < 0.05$), and values sharing a common letter are not significantly different ($p > 0.05$). (b) Time-course analysis of *NdTryp* knockdown efficiency after ds*NdTryp* injection. The result identified 24 h as the time point of maximal suppression, which was subsequently used for the bacterial challenge experiment. ***: $p < 0.001$, ****: $p < 0.0001$. (c) H&E staining of shrimp hepatopancreatic tissue, injection of ds*EGFP* following *V. parahaemolyticus* challenge for 72 **h**. (d) H&E staining of shrimp hepatopancreatic tissue, injection of ds*NdTryp* following *V. parahaemolyticus* challenge for 72 **h**. T: the star-shaped polygonal structure of the hepatopancreatic lumen; BM: basement membrane; B: secretory cells (B-cell); R: storage cells (R-cell); ECT: epithelial cells lining the tubules. Bars = 100 μm.

## 4.2. *NdTryp* expression and its biological significance

The exceptionally high expression of *NdTryp* in the hepatopancreas reflects the central role of this organ in both digestive and immune functions in crustaceans. This expression pattern is consistent with the dual functionality observed in our study. As the primary digestive organ, the hepatopancreas requires abundant proteolytic enzymes to process dietary proteins efficiently. Our ISH results revealed that *NdTryp* is predominantly distributed in R cells and ECTs (Fig 7b), suggesting that these cell types serve as the primary sites for its synthesis and secretion, thereby facilitating rapid mobilization during feeding. Trypsin has been implicated in embryonic development [9,35,36], although its specific role in crustacean

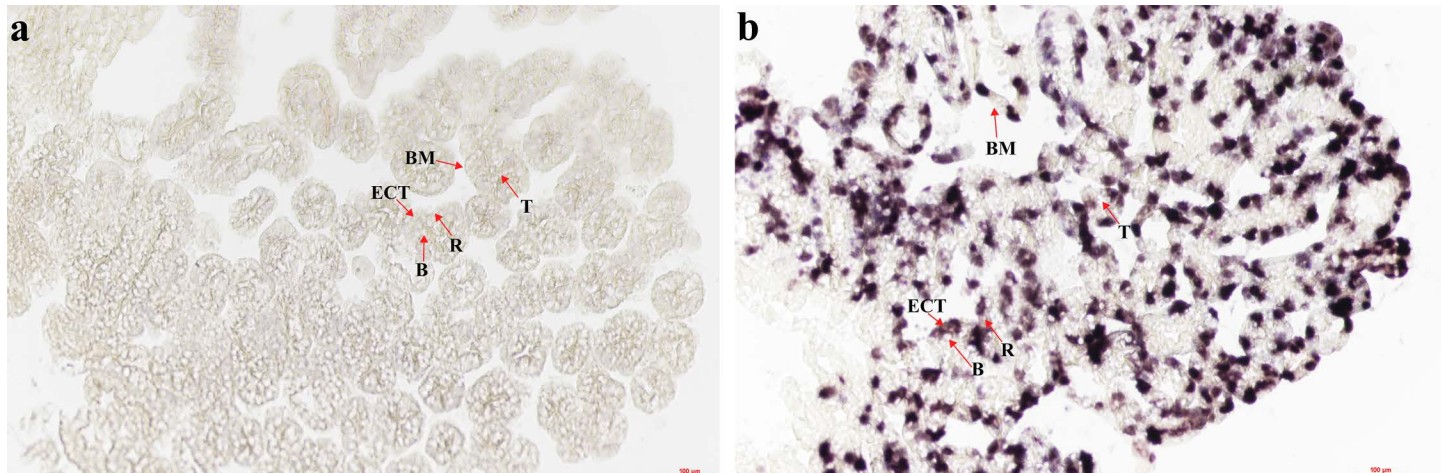

**Fig 7. Cellular localization of *NdTryp* mRNA in hepatopancreatic tissue.** T: the star-shaped polygonal structure of the hepatopancreatic lumen; BM: basement membrane; B: secretory cells (B-cell); R: storage cells (R-cell); ECT: epithelial cells lining the tubules. **(a)** The sense probe specific to *NdTryp* mRNA, subjected to BCIP/NBT staining, exhibited no alteration in coloration. Scale bars = 100 μm. **(b)** The antisense probe of *NdTryp* mRNA with BCIP/NBT staining is displayed in a bluish-violet color. Scale bar = 100 μm.

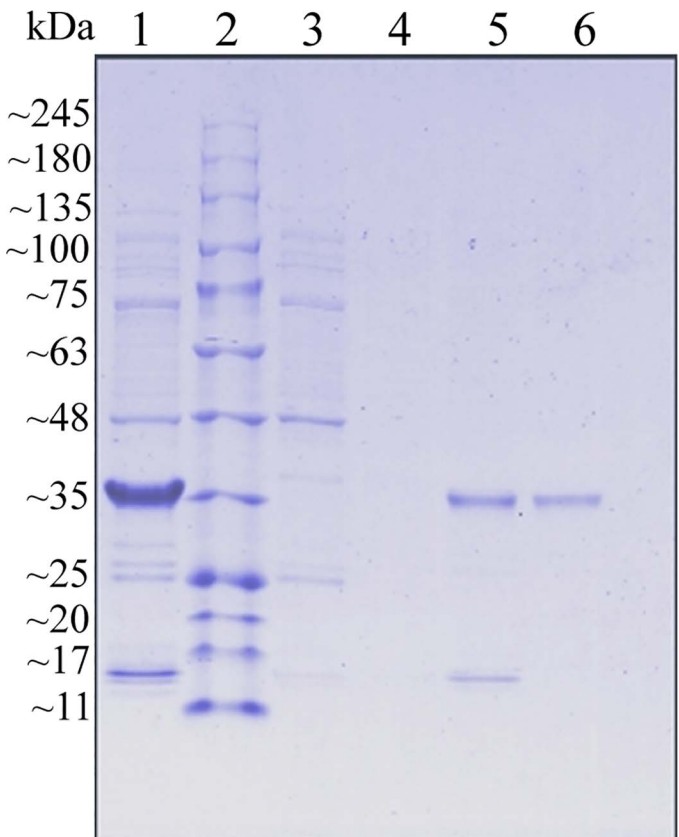

**Fig 8. SDS-PAGE analysis of recombinant protein.** Lane 1: Total proteins induced with IPTG. Lane 2: Protein marker. Lane 3-5: Protein bands of flow-through protein. Lane 6: Purified single rNdTryp protein band.

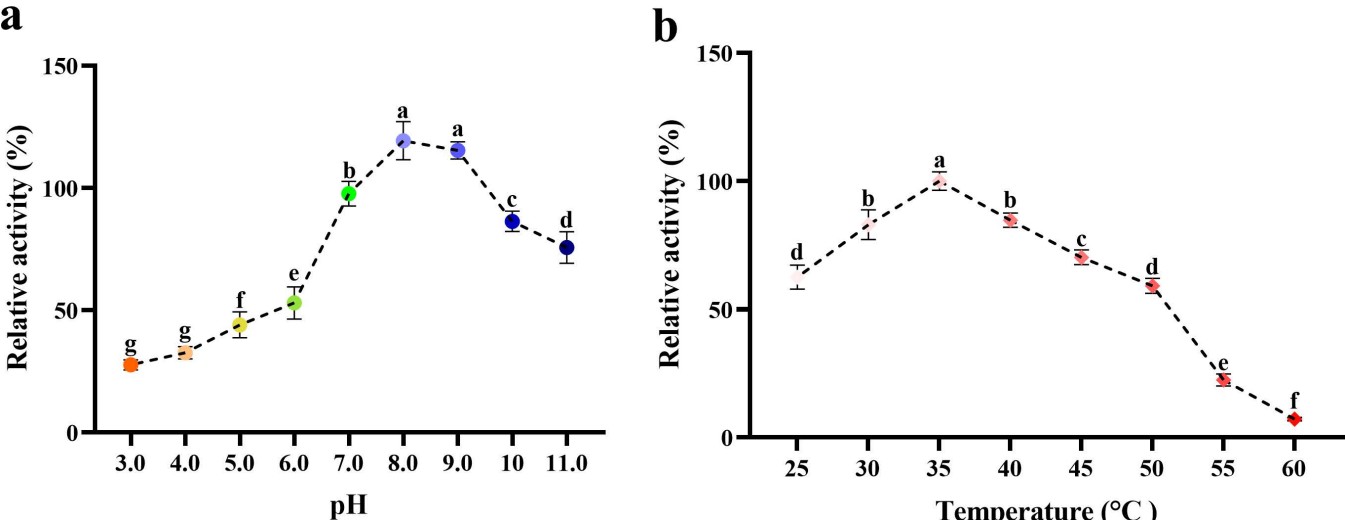

**Fig 9. Enzymatic characteristics of rNdTryp at different pH and temperature. (a)** The stability of different pH levels. The different letters (a, b, c, d, e, f, and g) on the lines designate statistical significance ($p<0.05$), and values sharing a common letter are not significantly different ($p>0.05$). **(b)** The stability of the different temperatures. The different letters (a, b, c, d, e, and f) on the lines designate statistical significance ($p<0.05$), and values sharing a common letter are not significantly different ($p>0.05$).

**Table 1. Effect of different metal ions on the relative activity of rNdTryp.**

| Metal ions (20 mM) | n | Relative activity (%) |
|---|---|---|
| $Ba^{2+}$ | 3 | 105±0.13 [cd] |
| $Ca^{2+}$ | 3 | 119±0.05 [bc] |
| $Cd^{2+}$ | 3 | 136±0.05 [a] |
| $Co^{2+}$ | 3 | 119±0.05 [bc] |
| $Cu^{2+}$ | 3 | 136±0.16 [a] |
| $Fe^{3+}$ | 3 | 29±0.11 [f] |
| $K^+$ | 3 | 86±0.04 [e] |
| $Li^+$ | 3 | 109±0.02 [bcd] |
| $Mg^{2+}$ | 3 | 98±0.05 [de] |
| $Mn^{2+}$ | 3 | 123±0.04 [ab] |

Note: The different lowercase English letters (a, b, c, d, e, f) designate statistical significance ($p<0.05$), and values sharing a common letter are not significantly different ($p>0.05$).

ontogenesis remains poorly defined. The present study reveals that *NdTryp* is not expressed early in the ontogenesis of *N. denticulata sinensis*, but maintains a high level of expression at later stages of embryonic development. During the early stages of the egg sac, the embryos will remain attached to the female shrimp and will need the yolk to provide all the energy and nutrients. Once hatched to the stage of free-swimming zoeae, they can catch their food [37]. During the hatching process, there is no significant yolk reserve in the juveniles of *Lysmata vittata*, indicating a high dependence on exogenous food after hatching [38]. Therefore, the high expression of *NdTryp* can rapidly transform into functional proteins, helping juvenile shrimp digest the food they capture. Given that shrimp embryos are enclosed in egg sacs [39], NdTryp may degrade the egg sacs to assist *N. denticulata sinensis* embryos in hatching successfully. These results indicate that the late embryonic stage of *N. denticulata sinensis* represents a crucial transitional window characterized by enhanced

protein metabolism, functional development of the digestive apparatus, and metabolic conditioning for hatching and autonomous survival. Beyond its canonical role in initiating the proteolytic cascade required for macronutrient digestion, trypsin also mediates cell signaling by cleaving protease-activated receptor 2 (PAR-2), a member of the G protein-coupled receptor family [40]. This activation of the G protein-coupled receptor governs diverse physiological processes, including secretory responses, vascular tone, and immune-inflammatory networks [41,42]. Collectively, our results demonstrate that NdTryp may play a critical role in multiple physiological processes during ontogenesis.

## 4.3. Proposed immune mechanisms

The hepatopancreas serves as both a digestive and a major immune organ in shrimp, responsible for synthesizing lectins, hemocyanin, proteases, and other immune-related molecules, while also being the primary target organ during *V. parahaemolyticus* [43]. *V. parahaemolyticus* challenge significantly upregulated *NdTryp* expression, whereas *NdTryp* knockdown resulted in extensive hepatopancreatic necrosis (Fig 6). Hepatopancreatic necrosis due to expression reduction of *NdTryp* exhibited a characteristic pattern of cellular disintegration, marked by progressive loss of cellular architecture and accumulation of sloughed necrotic debris within luminal spaces. These pathological changes mirror immunopathological signatures that occur when immune homeostasis is perturbed. Notably, *NdTryp* knockdown resulted in severe exacerbation of hepatopancreatic necrosis, providing compelling evidence for its essential role in maintaining tissue homeostasis and immune defense during bacterial infection. Although the precise mechanisms in *N. denticulata sinensis* remain incompletely understood, trypsin and related serine proteases participate in multiple innate immune pathways [44,45]. One possibility is that NdTryp functions as a convertase processing inactive propeptides into mature antimicrobial peptides (AMPs) — a maturation step analogous to defensin-5 processing in mammals [46]. Alternatively, as serine proteases, these enzymes can be integral to immune signaling cascades like the prophenoloxidase (proPO) system that mediates pathogen clearance [47, 48]. Serine proteases released from the hepatopancreas promote the conversion of proPPO to its active form, polyphenol oxidase (PPO) [49]. As a serine protease, trypsin may participate directly or indirectly in the cleavage and activation of proPO to phenoloxidase (PO). PO catalyzes the conversion of monophenols to quinones, which subsequently polymerize to form melanin, creating a barrier that restricts or eliminates pathogens [50]. The loss of host resistance and tissue integrity following *NdTryp* knockdown is therefore likely attributable to the disruption of these proteolytic pathways.

## 4.4. Biochemical properties and biotechnological potential

Based on the predictions, NdTryp was found to have a calcium-binding site, but it is unclear whether invertebrate trypsin requires $Ca^{2+}$ for maximum activity or stability [36]. In the present study, however, experiments on stability with metal ions revealed that a 20 mM $Ca^{2+}$ can enhance the activity of rNdTryp. $Fe^{3+}$ possesses a potent salting-out effect [51], suggesting that $Fe^{3+}$ inhibition may result from salting-out effects. Trypsin has various industrial applications, especially in the food industry, due to its high stability and activity [52]. This functional versatility is equally evident in aquaculture, where encapsulated proteases extracted from the hepatopancreas of *L. vannamei* have been successfully employed as feed additives to significantly enhance the digestibility and bioavailability of dietary nutrients [53]. The rNdTryp in this study demonstrated high enzymatic activity and maintained stability over a broad range of pH and temperature conditions. These findings provide a solid theoretical foundation for developing rNdTryp as a novel feed additive with the potential to simultaneously enhance digestive efficiency and strengthen host immunity. Given that *NdTryp* is highly expressed in R-cells and ECTs of hepatopancreas tissue, a theoretical approach could involve isolating these cells for trypsin production. However, this remains speculative, as crustacean cell culture systems are not yet well-established, and numerous technical challenges would need to be overcome before such an approach could be considered viable for industrial applications.

## 5. Conclusions

This study reports the first full-length trypsin cDNA from *N. denticulata sinensis*. *NdTryp* exhibited the highest expression in the hepatopancreas, localizing primarily to R-cells and ECTs. *NdTryp* was not expressed during early embryonic development, whereas its expression suddenly increased during late embryonic development. The mRNA transcripts increased after *V. parahaemolyticus* stimulation. Furthermore, inhibition of its expression can reduce the resistance of *N. denticulata sinensis* to *V. parahaemolyticus*, leading to partial necrosis of hepatopancreatic cells. The recombinant NdTryp (rNdTryp) exhibited a wide range of temperature and pH proteolytic activities, making it possible for rNdTryp to be a food waste degrader. Our results showed NdTryp as playing a critical role in shrimp embryonic development and antibacterial defense. The favorable enzymatic properties of the rNdTryp suggest its potential as a feed additive or biocatalyst. Realizing this potential, however, is contingent upon future studies designed to assess its industrial-scale feasibility and cost-effectiveness.

## Supporting information

**S1 File. Primers used in this study.**
(DOCX)

**S2 File. Genomic structure of NdTryp from *N. denticulata sinensis*.**
(DOCX)

**S3 File. NdTryp gene sequence.**
(DOCX)

**S4 File. Gel image of the ORF length of NdTryp amplified by NdTryp-ORF-F/R primers.**
(DOCX)

**S5 File. The efficiency of NdTryp-qPCR-F/R primers.**
(DOCX)

**S6 File. Gel image of NdTryp-qPCR-F/R primers was used to verify the specificity of qPCR primers.**
(DOCX)

**S7 File. Analysis of significant differences among different tissues.**
(DOCX)

**S8 File. High resolution Fig. 6c.**
(TIF)

**S9 File. High resolution Fig. 6d.**
(TIF)

**S10 File. High resolution Fig. 7a.**
(TIF)

**S11 File. High resolution Fig. 7b.**
(TIF)

**S12 File. The original image of Fig. 8.**
(TIF)

## Author contributions

**Conceptualization:** Yuying Sun.

**Formal analysis:** Dandan Feng.

**Funding acquisition:** Yuying Sun, Ji-Quan Zhang.

**Investigation:** Dandan Feng, Yakun Song, Zuqi Wu, Wuruo Liu, Yuting Pu, Yangcan Gao.

**Methodology:** Dandan Feng.

**Software:** Dandan Feng.

**Supervision:** Yuying Sun.

**Writing – original draft:** Dandan Feng.

**Writing – review & editing:** Dandan Feng, Yuying Sun, Ji-Quan Zhang.

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
