## [Decision Letter · Decision Letter 0]

10 Nov 2025

Dear Dr. Zhang,

Thank you for submitting your manuscript to PLOS ONE. After careful consideration, we feel that it has merit but does not fully meet PLOS ONE’s publication criteria as it currently stands. Therefore, we invite you to submit a revised version of the manuscript that addresses the points raised during the review process.

1. This manuscript not technically sound, and the data cannot support the conclusions. PLOS ONE is designed to communicate primary scientific research, and welcome submissions in any applied discipline that will contribute to the base of scientific knowledge. But this manuscript not adhere to the criteria for scientific research article that results show not sufficient to support the conclusion.

2. This manuscript has the statistical analysis problem.

3. This manuscript needs to adhere the PLOS Data Policy. The authors need to make all methods, materials and data underlying the findings in their manuscript fully available.

4. The whole manuscript is hard to read in current status. The language in submitted articles must be clear, correct, and unambiguous.

5. The revised manuscript needs to address each of the comments of the reviewers.

We look forward to receiving your revised manuscript.

Kind regards,

Tzong-Yueh Chen, Ph.D.

Academic Editor

PLOS ONE

Journal Requirements:

“This work was supported by the National Natural Science Foundation of China (Grant Nos. 32172954, 32373121) and the Natural Science Foundation of Hebei Province of China (D2023201002).”

5. In the online submission form you indicate that your data is not available for proprietary reasons and have provided a contact point for accessing this data. Please note that your current contact point is a co-author on this manuscript. According to our Data Policy, the contact point must not be an author on the manuscript and must be an institutional contact, ideally not an individual. Please revise your data statement to a non-author institutional point of contact, such as a data access or ethics committee, and send this to us via return email. Please also include contact information for the third party organization, and please include the full citation of where the data can be found.

6. PLOS requires an ORCID iD for the corresponding author in Editorial Manager on papers submitted after December 6th, 2016. Please ensure that you have an ORCID iD and that it is validated in Editorial Manager. To do this, go to ‘Update my Information’ (in the upper left-hand corner of the main menu), and click on the Fetch/Validate link next to the ORCID field. This will take you to the ORCID site and allow you to create a new iD or authenticate a pre-existing iD in Editorial Manager.

7. Your ethics statement should only appear in the Methods section of your manuscript. If your ethics statement is written in any section besides the Methods, please move it to the Methods section and delete it from any other section. Please ensure that your ethics statement is included in your manuscript, as the ethics statement entered into the online submission form will not be published alongside your manuscript.

8. Please include captions for your Supporting Information files at the end of your manuscript, and update any in-text citations to match accordingly. Please see our Supporting Information guidelines for more information: http://journals.plos.org/plosone/s/supporting-information .

9. PLOS ONE now requires that authors provide the original uncropped and unadjusted images underlying all blot or gel results reported in a submission’s figures or Supporting Information files. This policy and the journal’s other requirements for blot/gel reporting and figure preparation are described in detail at https://journals.plos.org/plosone/s/figures#loc-blot-and-gel-reporting-requirements and https://journals.plos.org/plosone/s/figures#loc-preparing-figures-from-image-files. When you submit your revised manuscript, please ensure that your figures adhere fully to these guidelines and provide the original underlying images for all blot or gel data reported in your submission. See the following link for instructions on providing the original image data: https://journals.plos.org/plosone/s/figures#loc-original-images-for-blots-and-gels.

Reviewers' comments:

Reviewer's Responses to Questions

**Comments to the Author**

1. Is the manuscript technically sound, and do the data support the conclusions?

Reviewer #1: Yes

Reviewer #2: Partly

2. Has the statistical analysis been performed appropriately and rigorously?

Reviewer #1: I Don't Know

Reviewer #2: No

3. Have the authors made all data underlying the findings in their manuscript fully available?

Reviewer #1: No

Reviewer #2: Yes

4. Is the manuscript presented in an intelligible fashion and written in standard English?

Reviewer #1: No

Reviewer #2: No

Reviewer #1: 1. Data availability does not comply with PLOS ONE policy.

Your current statement that “data are available upon reasonable request” is not sufficient. All underlying data should be made publicly accessible via a repository or included in the manuscript/Supplementary Information. Please revise accordingly.

2. English usage requires substantial improvement.

The manuscript contains numerous instances of repetitive or awkward sentence structures, including excessive passive voice. A thorough language editing by a native English speaker or a professional service is strongly advised.

3. Statistical details in figures are insufficient.

Figures such as Fig. 5 and Fig. 6 lack proper annotation of statistical significance (e.g., a/b/c labelling, p-values). Please include exact sample sizes (n), error bars (mean ± SD or SEM), and statistical tests used in each figure legend.

4. Visual presentation of figures requires enhancement.

Images in Fig. 6 and Fig. 7 (H&E and ISH) lack scale bars and cellular annotations. Please ensure all micrographs are labelled clearly with appropriate magnification and cell types.

5. Potential mechanistic insight is underdeveloped.

While you report that NdTryp may be involved in immune defence against Vibrio parahaemolyticus, the molecular mechanism remains speculative. Please elaborate on possible pathways or propose a model, even if hypothetical.

6. Discussion contains overextended claims.

Statements regarding industrial application for food waste degradation or environmental protection should be toned down or qualified, as current evidence is preliminary.

7. Ethical statements are acceptable but could be clarified.

Although no vertebrate animals were used, please state explicitly whether all procedures were approved by institutional guidelines and whether ARRIVE guidelines were followed.

8. Literature references need updating and better integration.

Some cited references are outdated (e.g., 1990s), and others are not effectively linked to claims made in the text. Please ensure recent and relevant studies are cited appropriately.

9. Clarify biological replicates and experimental repeats.

While some methods mention “three replicates,” it is unclear whether these are technical or biological replicates. Please clarify replication strategy for each major experiment.

Reviewer #2: Reviewer’s comments

Title: Functional analysis of trypsin from Neocaridina denticulata sinensis: Cellular localization, ontogenesis, immune stimulation, and enzymatic characteristics

Manuscript Number: PONE-D-25-30024

Journal: PLOS ONE

The research work entitled “Functional analysis of trypsin from Neocaridina denticulata sinensis: Cellular localization, ontogenesis, immune stimulation, and enzymatic characteristics” is the first to reveal the molecular properties of trypsin in N. denticulata sinensis and its potential association with immune functions. The findings hold significant social and industrial implications: on one hand, they provide a theoretical basis for extending the shelf life of shrimp products and improving issues related to texture softening; on the other hand, understanding the role of endogenous enzymes in defense responses contributes to the development of antibiotic-free disease prevention strategies for shrimp, promoting environmentally friendly, safe, and sustainable aquaculture. However, this study still has certain shortcomings. For instance, the image resolution is too low to meet publication standards, and the overall structure of the manuscript appears somewhat disorganized, with arguments that could benefit from clearer logical flow. If the authors can improve image quality, supplement experimental data, and reorganize the section structure, the readability and academic value of the paper would be greatly enhanced. Therefore, I recommend a “major revision” for this manuscript in PLOS ONE. Specific comments are provided below:

Specific comments

1. The English writing of the manuscript is recommended to undergo professional language editing.

2. The title could be refined to more precisely reflect the core aspect of the study, such as its physiological characterization.

3. The current Graphical Abstract could be better aligned with the manuscript’s main theme and may benefit from being redrawn with brief explanatory text or icon labels to enhance clarity.

Abstract: The abstract is somewhat lengthy and could benefit from a clearer structure to highlight the main focus.

1. The organization of the abstract is somewhat unclear, with excessive detail that blurs the distinction between the background, objectives, methods, and conclusions. Reorganizing the structure to present these elements more concisely and clearly is recommended.

Introduction: The structure appears somewhat unclear, and the main points could be better emphasized.

1. A clearer and more focused organization in the introduction would greatly enhance overall readability. The current content appears somewhat disordered, which may hinder reader comprehension. It is recommended to condense the paragraphs and focus on presenting essential background information and the main research motivation.

2. The explanation of trypsin’s physiological roles could be streamlined, as similar descriptions appear in both the Abstract and Introduction

3. The manuscript currently addresses several broad topics—including digestive enzyme functions, antibiotic resistance, food waste utilization, and genetic engineering—which makes the overall theme somewhat diffuse. It is suggested to narrow the focus and emphasize the research motivation related to NdTryp’s physiological and immune functions, while discussing the applied aspects (e.g., food waste and enzyme recycling) later in the mauescript.

Materials and methods: Inconsistent product source labeling.

1. It is recommended to standardize the format used for indicating product sources, as some product descriptions currently lack consistency. For example: glass slides

(CITOTEST, 188105)、methanol (MACKLIN, Shanghai, China)。

Result: Revisions required.

1. The resolution of all figures is insufficient and does not meet the journal’s publication standards (https://journals.plos.org/plosone/s/figures)

2. In Fig. 1, the color markings for the poly(A) tails (901–906 bp) and the underlined region (879–900 bp) are inconsistent; additionally, the underlined region for the Trp_SPc domain does not match the description in the text (text states 236 aa, while the figure shows 232 aa)

3. In Fig. 5a, the comparison baseline is unclear. Moreover, describing NdTryp mRNA expression levels across different tissues as “at the same level” is inaccurate, as noticeable differences exist (e.g., in the Te tissue).

4. The logical connection and descriptive explanation between the subfigures in Fig. 6 need to be strengthened, especially for Fig. 6b.

5. The figure legends for Fig. 5, Fig. 6(a), and Fig. 9 lack explanations for the letter labels a-g shown on the graphs.

6. The legend for Table 1 is missing explanations for the symbols a-f.

Discussion: Partial revision required

1. Lines 293–294: The statement “Given that shrimp…successfully.” appears speculative and could be strengthened by adding appropriate references to support its credibility.

2. Lines 300–302: In the sentence “From the…against V. parahaemolyticusit.” the proposed concept could be clarified further by providing a clearer explanation of the underlying mechanism and supporting it with relevant references.

3. Lines 307–314: This paragraph discusses potential applications. It is recommended that the authors first complete the explanation of the trypsin-related mechanisms before introducing this section. Additionally, the logical connection between paragraphs should be strengthened to improve flow and coherence.

**Do you want your identity to be public for this peer review?** For information about this choice, including consent withdrawal, please see our Privacy Policy

Reviewer #1: No

Reviewer #2: No

---

## [Author Response · Author response to Decision Letter 1]

17 Nov 2025

Detailed Response to Reviewers and Journal Requirements

Dear Editor,

Thank you for your useful comments and suggestions on our manuscript (PONE-D-25-30024). We have tried our best to revise the manuscript according to the reviewers’ suggestions, and detailed corrections are listed below point by point:

Journal Requirements:

Question 1. Please ensure that your manuscript meets PLOS ONE's style requirements, including those for file naming. The PLOS ONE style templates can be found at

Response 1. Thank you for your email and the reminder regarding the journal’s formatting requirements. We have carefully reviewed the PLOS ONE's style templates for the main body and the title/authors/affiliations sections. We have revised our manuscript accordingly to ensure it fully complies with all the specified requirements, including file naming.

Question 2. Please note that funding information should not appear in any section or other areas of your manuscript. We will only publish funding information present in the Funding Statement section of the online submission form. Please remove any funding-related text from the manuscript.

Response 2. Any text related to funding has been removed from the revised manuscript as requested.

Question 3. We note that the grant information you provided in the ‘Funding Information’ and ‘Financial Disclosure’ sections do not match.

Response 3: We have carefully reviewed and corrected the manuscript. The grant numbers in the ‘Funding Information’ section are now accurate and consistent with the ‘Financial Disclosure’ section, as you requested.

Question 4. Thank you for stating the following financial disclosure:

“This work was supported by the National Natural Science Foundation of China (Grant Nos. 32172954, 32373121) and the Natural Science Foundation of Hebei Province of China (D2023201002).”

Response 4: As requested, we have now added the role of the funders to the ‘Funding’ section of our revised manuscript. The amended statement is as follows: The funders (Jiquan Zhang) had a role in the study design, data collection, and analysis, as well as the decision to publish. We have also included this statement in our revised cover letter.

Question 5. In the online submission form you indicate that your data is not available for proprietary reasons and have provided a contact point for accessing this data. Please note that your current contact point is a co-author on this manuscript. According to our Data Policy, the contact point must not be an author on the manuscript and must be an institutional contact, ideally not an individual. Please revise your data statement to a non-author institutional point of contact, such as a data access or ethics committee, and send this to us via return email. Please also include contact information for the third party organization, and please include the full citation of where the data can be found.

Response 5: As our data is part of an ongoing study, it cannot be deposited in a public repository at this time. However, according to your instructions, we have included all the basic data of this study as supporting information files.

Question 6. PLOS requires an ORCID iD for the corresponding author in Editorial Manager on papers submitted after December 6th, 2016. Please ensure that you have an ORCID iD and that it is validated in Editorial Manager. To do this, go to ‘Update my Information’ (in the upper left-hand corner of the main menu), and click on the Fetch/Validate link next to the ORCID field. This will take you to the ORCID site and allow you to create a new iD or authenticate a pre-existing iD in Editorial Manager.

Response 6: Thank you for reminding us about the ORCID iD requirement. I have now validated my ORCID iD in the Editorial Manager system as requested. The ORCID iD (0000-0001-5063-339x) has been successfully linked to my account through the ‘Update my Information’ section.

Question 7. Your ethics statement should only appear in the Methods section of your manuscript. If your ethics statement is written in any section besides the Methods, please move it to the Methods section and delete it from any other section. Please ensure that your ethics statement is included in your manuscript, as the ethics statement entered into the online submission form will not be published alongside your manuscript.

Response 7: Thank you for your guidance regarding the ethics statement placement. Following your instructions, we have reviewed our manuscript and confirmed that the ethics statement now appears only in the Materials and methods section (Section 2.1). In addition, we have removed any duplicate ethics statements from other sections of the manuscript.

Question 8. Please include captions for your Supporting Information files at the end of your manuscript, and update any in-text citations to match accordingly. Please see our Supporting Information guidelines for more information: http://journals.plos.org/plosone/s/supporting-information.

Response 8: Thank you for your guidance regarding the Supporting Information files. Following your instructions, we have now included captions for all Supporting Information files at the end of our manuscript. We have also updated all in-text citations to ensure they match the Supporting Information captions accordingly.

Question 9. PLOS ONE now requires that authors provide the original uncropped and unadjusted images underlying all blot or gel results reported in a submission’s figures or Supporting Information files. This policy and the journal’s other requirements for blot/gel reporting and figure preparation are described in detail at https://journals.plos.org/plosone/s/figures#loc-blot-and-gel-reporting-requirements and https://journals.plos.org/plosone/s/figures#loc-preparing-figures-from-image-files. When you submit your revised manuscript, please ensure that your figures adhere fully to these guidelines and provide the original underlying images for all blot or gel data reported in your submission. See the following link for instructions on providing the original image data: https://journals.plos.org/plosone/s/figures#loc-original-images-for-blots-and-gels.

Response 9: Thank you for your guidance regarding the original image requirements for blot and gel data. We have carefully reviewed the journal’s guidelines and have provided the original, unadjusted underlying image for the SDS-PAGE gel reported in Fig. 8. This raw image has been included as a Supporting Information file (S8 File), and a corresponding caption has been added at the end of the manuscript as requested. The original full-size gel also contained lanes from a separate, unrelated study, and was therefore physically cut before imaging. The image we have provided is the complete and unadjusted original for all lanes pertaining to this manuscript.

Question 10. If the reviewer comments include a recommendation to cite specific previously published works, please review and evaluate these publications to determine whether they are relevant and should be cited. There is no requirement to cite these works unless the editor has indicated otherwise.

Response 10: We sincerely thank the editor for this insightful and constructive comment. We have carefully reviewed and evaluated the cited references to determine their relevance to this study.

Reviewers’ comments

Reviewer #1, Comments to PONE-D-25-30024:

Question 1. Data availability does not comply with PLOS ONE policy.

Your current statement that “data are available upon reasonable request” is not sufficient. All underlying data should be made publicly accessible via a repository or included in the manuscript/Supplementary Information. Please revise accordingly.

Response 1: Thank you for your guidance regarding the Data Availability Statement. We have revised the manuscript accordingly. All underlying data have now been included as Supporting Information files, and the Data Availability Statement has been updated to reflect this change.

Question 2. English usage requires substantial improvement.

The manuscript contains numerous instances of repetitive or awkward sentence structures, including excessive passive voice. A thorough language editing by a native English speaker or a professional service is strongly advised.

Response 2: Thanks for the reviewer’s advice. We read the paper carefully and invited an Indian professor to help modify the language.

Question 3. Statistical details in figures are insufficient.

Figures such as Fig. 5 and Fig. 6 lack proper annotation of statistical significance (e.g., a/b/c labelling, p-values). Please include exact sample sizes (n), error bars (mean ± SD or SEM), and statistical tests used in each figure legend.

Response 3: We sincerely thank the reviewer for constructive comments. We have annotated the statistical significance in the legends of the figures. The exact sample size (n), error bars (mean ± SD), and statistical tests were added in the Materials and methods section.

Question 4. Visual presentation of figures requires enhancement.

Images in Fig. 6 and Fig. 7 (H&E and ISH) lack scale bars and cellular annotations. Please ensure all micrographs are labelled clearly with appropriate magnification and cell types.

Response 4: Thank you for your valuable feedback regarding the visual presentation of our figures. The issue was caused by the submission system’s file size limitations, which required us to compress the images. This, unfortunately, resulted in a loss of detail, making the scale bars and cellular annotations difficult to read. Our original, high-resolution micrographs do contain clear scale bars and annotations. To address this, we have taken the following steps: We have uploaded the original, high-resolution versions of Fig. 6 and Fig. 7 as a new Supporting Information file named ‘S4-S7 File: High resolution Fig. 6 and Fig. 7’. These images clearly show all the required scale bars and cellular annotations (e.g., B-cells, R-cells, ECTs).

Question 5. Potential mechanistic insight is underdeveloped.

While you report that NdTryp may be involved in immune defence against Vibrio parahaemolyticus, the molecular mechanism remains speculative. Please elaborate on possible pathways or propose a model, even if hypothetical.

Response 5: We sincerely thank the reviewer for this insightful and constructive comment. Based on your suggestion, we have added a detailed explanation and hypothetical model of the immune pathways that NdTryp may be involved in in the Discussion section of the revised manuscript to provide deeper mechanistic insights. Please refer to page 11, lines 311-315 of the revised manuscript for details.

Question 6. Discussion contains overextended claims.

Statements regarding industrial application for food waste degradation or environmental protection should be toned down or qualified, as current evidence is preliminary.

Response 6: We sincerely thank the reviewer for this insightful and constructive comment. Based on the reviewer’s suggestion, we have carefully revised the Discussion and Conclusions sections, weakening and limiting the relevant statements to make them more rigorous and in line with existing evidence. Please refer to page 12, lines 316-326, and page 13, lines 347-350 of the revised manuscript for details.

Question 7: Ethical statements are acceptable but could be clarified.

Although no vertebrate animals were used, please state explicitly whether all procedures were approved by institutional guidelines and whether ARRIVE guidelines were followed.

Response 7: Thank you for your feedback and the suggestion to clarify our ethical statement. We agree that a more explicit statement enhances the transparency of our work. Following your recommendation, we have revised and consolidated the ethical statement in the Methods section of our manuscript. The revised statement now explicitly confirms that all procedures were approved by institutional guidelines and that the ARRIVE guidelines were followed. To improve clarity, we have revised the relevant section in the manuscript (Section 2.1, pages 3-4) to explicitly state these points.

Question 8: Literature references need updating and better integration.

Some cited references are outdated (e.g., 1990s), and others are not effectively linked to claims made in the text. Please ensure recent and relevant studies are cited appropriately.

Response 8: Thank you for your valuable comment regarding the literature references in our manuscript. We acknowledge that some references from the 1990s are cited in our manuscript. However, we would like to respectfully clarify that these older references represent seminal and foundational works in the field of crustacean trypsin research. In addition, we conducted a comprehensive search and review of recent literature. The literature with strong relevance has been added to the revised manuscript, while the literature with low relevance has been removed from the citation list. Please refer to the References section for detailed information.

Question 9. Clarify biological replicates and experimental repeats.

While some methods mention “three replicates,” it is unclear whether these are technical or biological replicates. Please clarify replication strategy for each major experiment.

Response 9: Thank you for your valuable feedback regarding the clarification of our experimental replicates. We have now revised the Materials and Methods section to explicitly differentiate between biological and technical replicates for each major experiment. Please refer to page 5, line 121, lines 129-130, lines 140-141; and page 7, lines 198-200 of the revised manuscript for details.

Reviewers’ comments

Reviewer #2, Comments to PONE-D-25-30024:

Reviewer #2: The research work entitled “Functional analysis of trypsin from Neocaridina denticulata sinensis: Cellular localization, ontogenesis, immune stimulation, and enzymatic characteristics” is the first to reveal the molecular properties of trypsin in N. denticulata sinensis and its potential association with immune functions. The findings hold significant social and industrial implications: on one hand, they provide a theoretical basis for extending the shelf life of shrimp products and improving issues related to texture softening; on the other hand, understanding the role of endogenous enzymes in defense responses contributes to the development of antibiotic-free disease prevention strategies for shrimp, promoting environmentally friendly, safe, and sustainable aquaculture. However, this study still has certain shortcomings. For instance, the image resolution is too low to meet publication standards, and the overall structure of the manuscript appears somewhat disorganized, with arguments that could benefit from clearer logical flow. If the authors can improve image quality, supplement experimental data, and reorganize the section structure, the readability and academic value of the paper would be greatly enhanced. Therefore, I recommend a “major revision” for this manuscript in PLOS ONE. Specific comments are provided below:

Spec

---

## [Decision Letter · Decision Letter 1]

10 Dec 2025

Dear Dr. Zhang,

Thank you for submitting your manuscript to PLOS ONE. After careful consideration, we feel that it has merit but does not fully meet PLOS ONE’s publication criteria as it currently stands. Therefore, we invite you to submit a revised version of the manuscript that addresses the points raised during the review process.

1. This manuscript not technically sound, and the data cannot support the conclusions. PLOS ONE is designed to communicate primary scientific research, and welcome submissions in any applied discipline that will contribute to the base of scientific knowledge. But this manuscript not adhere to the criteria for scientific research article that results show not sufficient to support the conclusion.

2. This manuscript has the statistical analysis problem.

3. The whole manuscript is hard to read in current status. The language in submitted articles must be clear, correct, and unambiguous.

We look forward to receiving your revised manuscript.

Kind regards,

Tzong-Yueh Chen, Ph.D.

Academic Editor

PLOS One

Journal Requirements:

Reviewers' comments:

Reviewer's Responses to Questions

**Comments to the Author**

Reviewer #1: All comments have been addressed

Reviewer #2: (No Response)

2. Is the manuscript technically sound, and do the data support the conclusions?

Reviewer #1: Yes

Reviewer #2: Partly

3. Has the statistical analysis been performed appropriately and rigorously?

Reviewer #1: Yes

Reviewer #2: N/A

4. Have the authors made all data underlying the findings in their manuscript fully available?

Reviewer #1: Yes

Reviewer #2: Yes

5. Is the manuscript presented in an intelligible fashion and written in standard English?

Reviewer #1: Yes

Reviewer #2: No

Reviewer #1: This manuscript has been fully revised in accordance with the reviewers’ comments and is now suitable for publication.

Reviewer #2: The research work entitled “Physiological and biochemical characterization of trypsin from Neocaridina denticulata sinensis and its roles in ontogenesis and immune response”

presents interesting data on the characterization of NdTryp and addresses a topic of potential relevance to crustacean physiology and immunity. However, several aspects of the writing and organization currently limit the clarity and overall impact of the work. Across multiple sections, the narrative tends to lose focus, with frequent inclusion of background information or speculative interpretations that extend beyond the scope of the presented data. As a result, the central findings are sometimes overshadowed, and the connection between the study’s aims, methods, results, and broader significance becomes less clear. Substantial revisions to the structure, conciseness, and alignment of each section with the core objectives of the study would greatly improve readability and help strengthen the manuscript’s scientific message. In addition, several sentences would benefit from careful English editing to enhance clarity, precision, and overall fluency. Therefore, I recommend a “minor revision” for this manuscript in PLOS ONE. Specific comments are provided below:

Specific comments

This manuscript remains linguistically unqualified for publication. It is strongly recommended to have the entire text thoroughly revised by a professional native English editor.

Abstract:

1. The abstract reads as a list of results rather than a cohesive summary. The study rationale and knowledge gap are insufficiently stated.

2. Several experimental details (e.g., specific metal ions) are too fine-grained for an abstract and should be summarized.

3. The logical flow is fragmented; developmental data, immune response, and biochemical assays appear disconnected.

Suggestions for Restructuring:

1. Begin with a concise background that clearly states the research gap.

2. Follow with a single sentence summarizing the study aim.

3. Group major findings into biological (expression/immune) and biochemical (activity) results.

4. End with a clear, complete significance statement relevant to crustacean biology or aquaculture.

Introduction: A more concise and structured presentation will better guide readers toward the specific aims of the study.

1. Certain sections provide extensive textbook-level descriptions and broader contextual information (e.g., food waste, CO₂ emissions, industrial proteases) that may not directly support the central theme of the study. Streamlining these areas would help maintain focus.

2. Background information relevant to this species appears somewhat late in the Introduction and might be more effective if moved forward.

3. The knowledge gap is not clearly stated. It is unclear what is unknown about trypsin in N. denticulata sinensis, and how this study fills that gap.

Materials and methods: Partial revision required

1. The distinction between biological and technical replicates could be described more clearly across experiments.

2. The reported dsRNA concentration seems unusually high (1,000 μg/μL); confirming this value would strengthen confidence in the RNAi experiments.

Result: The logical flow is acceptable; only English language editing is required.

Discussion: Partial revision required

A more targeted discussion will help emphasize the contributions of this study and align them closely with the presented data.

1. The flow of ideas occasionally shifts between distant topics, and organizing the discussion around the key results (developmental expression, immune response, biochemical properties) may improve clarity.

2. Certain important findings, such as the expression pattern and RNAi phenotypes, could be discussed in greater depth with regard to their biological relevance.

**Do you want your identity to be public for this peer review?** For information about this choice, including consent withdrawal, please see our Privacy Policy

Reviewer #1: No

Reviewer #2: No

---

## [Author Response · Author response to Decision Letter 2]

17 Dec 2025

Detailed Response to Reviewers

Dear Editor,

Thank you for your useful comments and suggestions on our manuscript (PONE-D-25-30024R1). We have tried our best to revise the manuscript according to the reviewers’ suggestions, and detailed corrections are listed below point by point:

Reviewers’ comments

Reviewer #1: This manuscript has been fully revised in accordance with the reviewers’ comments and is now suitable for publication.

Response to Reviewer #1: We are very pleased to learn that you are satisfied with our revisions and agree that all previous concerns have been addressed. We are sincerely grateful for your time and effort in this process. Your insightful and constructive feedback has been invaluable and has significantly improved the quality and clarity of our manuscript.

Reviewers’ comments

Title: Physiological and biochemical characterization of trypsin from Neocaridina denticulata sinensis and its roles in ontogenesis and immune response

Manuscript Number: PONE-D-25-30024R1

Journal: PLOS ONE

The research work entitled “Physiological and biochemical characterization of trypsin from Neocaridina denticulata sinensis and its roles in ontogenesis and immune response” presents interesting data on the characterization of NdTryp and addresses a topic of potential relevance to crustacean physiology and immunity. However, several aspects of the writing and organization currently limit the clarity and overall impact of the work. Across multiple sections, the narrative tends to lose focus, with frequent inclusion of background information or speculative interpretations that extend beyond the scope of the presented data. As a result, the central findings are sometimes overshadowed, and the connection between the study’s aims, methods, results, and broader significance becomes less clear. Substantial revisions to the structure, conciseness, and alignment of each section with the core objectives of the study would greatly improve readability and help strengthen the manuscript’s scientific message. In addition, several sentences would benefit from careful English editing to enhance clarity, precision, and overall fluency. Therefore, I recommend a “minor revision” for this manuscript in PLOS ONE. Specific comments are provided below:

Question: Specific comments

This manuscript remains linguistically unqualified for publication. It is strongly recommended to have the entire text thoroughly revised by a professional native English editor.

Response: Thank you for your comment. We fully agree that the English language needs substantial improvement. The entire manuscript (including the abstract and main text) has now been thoroughly revised by a professional native English editor to improve grammar, clarity, readability, and academic style. We have also carefully checked terminology consistency throughout the manuscript.

Abstract:

Question 1. The abstract reads as a list of results rather than a cohesive summary. The study rationale and knowledge gap are insufficiently stated.

Response 1: We thank the reviewer for this valuable suggestion. We have further refined the abstract to enhance its clarity, logical flow, and scientific rigor while maintaining its comprehensive coverage of our key findings. The modifications specifically address:

1) improved logical progression from background to objectives to results.

2) enhanced quantitative precision in reporting key findings.

3) strengthened emphasis on the novelty and implications of our work.

4) Better integration of functional and biotechnological aspects.

We believe these refinements have strengthened the manuscript’s presentation and accessibility to a broader readership.

Question 2. Several experimental details (e.g., specific metal ions) are too fine-grained for an abstract and should be summarized.

Response 2: We thank the reviewer for this valuable suggestion. Therefore, we revised the Abstract by:

1) removing the enumeration of individual ions and associated quantitative effects.

2) summarizing this section as “activity was modulated by divalent/trivalent metal ions,” while retaining the key take-home message that rNdTryp shows broad pH/temperature tolerance and ion-dependent modulation.

Question 3. The logical flow is fragmented; developmental data, immune response, and biochemical assays appear disconnected.

Suggestions for Restructuring:

1. Begin with a concise background that clearly states the research gap.

2. Follow with a single sentence summarizing the study aim.

3. Group major findings into biological (expression/immune) and biochemical (activity) results.

4. End with a clear, complete significance statement relevant to crustacean biology or aquaculture.

Response 3: We thank the reviewer for this insightful suggestion. We have revised the Abstract to improve coherence and ensure a single, continuous logic chain.

1) The background/research gap was moved to the opening sentence to clarify what is unknown regarding NdTryp across development, immunity, and enzymatic traits.

2) We added one explicit aim sentence to unify the experimental modules (gene characterization, biological function, and recombinant biochemical profiling).

3) We grouped the major findings into two clearly labeled blocks.

4) We rewrote the final sentence as a complete significance statement highlighting relevance to crustacean biology and aquaculture/biotechnology potential.

Introduction: A more concise and structured presentation will better guide readers toward the specific aims of the study.

Question 1. Certain sections provide extensive textbook-level descriptions and broader contextual information (e.g., food waste, CO₂ emissions, industrial proteases) that may not directly support the central theme of the study. Streamlining these areas would help maintain focus.

Response 1: Thank you for your feedback and the suggestion. The corresponding parts of the manuscript have been rewritten, removing extensive textbook-level descriptions and broader contextual information such as food waste, CO₂ emissions, and industrial proteases.

Question 2. Background information relevant to this species appears somewhat late in the Introduction and might be more effective if moved forward.

Response 2: Thank you for this helpful suggestion. We have revised the Introduction by moving the background information relevant to this species to an earlier position and updating the relevant references accordingly, so that the rationale and context are established upfront (see revised Introduction section).

Question 3. The knowledge gap is not clearly stated. It is unclear what is unknown about trypsin in N. denticulata sinensis, and how this study fills that gap.

Response 3: We sincerely thank the reviewer for this constructive comment. We agree that the knowledge gap in the original manuscript was insufficiently articulated. To address this concern, we have substantially revised the Introduction to explicitly state what remains unknown about trypsin in Neocaridina denticulata sinensis and how our study systematically fills these gaps. Please refer to page 3, lines 64-67 of the revised manuscript for details.

Materials and methods: Partial revision required

Question 1. The distinction between biological and technical replicates could be described more clearly across experiments.

Response 1: We thank the reviewer for raising this important point regarding experimental replicates. We have carefully reviewed our manuscript and provided clarifications as follows:

1. Biological Replicates

For tissue expression analysis (Section 2.6)

For bacterial challenge experiments (Section 2.5)

For the embryonic development study (Section 2.6)

2. Technical Replicates

For qPCR assays (Section 2.6)

For enzymatic characterization (Section 2.10)

3. Statistical Analysis Clarification

Statistical analysis (Section 2.11)

Question 2. The reported dsRNA concentration seems unusually high (1,000 μg/μL); confirming this value would strengthen confidence in the RNAi experiments.

Response 2: Thank you for pointing this out. The dsRNA concentration “1,000 μg/μL” was a Unit typographical error. The correct concentration is 1,000 ng/μL (i.e., 1 μg/μL). We have corrected the unit throughout the manuscript (2.4 Methodology of RNA interference (RNAi) section, page 5, lines 113–114). This correction does not affect the RNAi experimental design, dosage, or the conclusions of the study.

Question Result: The logical flow is acceptable; only English language editing is required.

Response: Thank you for this helpful suggestion. We have carefully revised the English language throughout the manuscript with the assistance of a native English-speaking colleague. The revisions addressed grammar, sentence clarity, and consistency in scientific expression. We believe the revised manuscript now meets the journal’s language standards.

Discussion: Partial revision required

A more targeted discussion will help emphasize the contributions of this study and align them closely with the presented data.

Question 1. The flow of ideas occasionally shifts between distant topics, and organizing the discussion around the key results (developmental expression, immune response, biochemical properties) may improve clarity.

Response 1: Thank you for this constructive suggestion regarding the organization and flow of the Discussion section. To improve the logical structure, our key findings would enhance manuscript clarity. We have completely reorganized the Discussion section into four clearly delineated subsections:

1) Section 4.1 - Structural features and evolutionary conservation of NdTryp: We now begin with the molecular foundation, discussing the catalytic triad, calcium-binding sites, and β-barrel architecture that underpin biochemical properties of NdTryp.

2) Section 4.2 - NdTryp expression and its biological significance: This section provides a focused analysis of the developmental expression pattern, interpreting the biphasic profile in the context of yolk dependence, exogenous feeding transition, and potential hatching facilitation.

3) Section 4.3 - Proposed immune mechanisms: We consolidated all immune-related findings here, including:

Expression dynamics following V. parahaemolyticus challenge;

RNAi phenotype description and interpretation;

Proposed mechanistic pathways;

Integration of histopathological observations with molecular mechanisms.

4) Section 4.4 - Biochemical properties and biotechnological potential: This section synthesizes enzymatic characterization data with practical applications, concluding with cellular localization findings and production feasibility considerations.

Question 2. Certain important findings, such as the expression pattern and RNAi phenotypes, could be discussed in greater depth with regard to their biological relevance.

Response 2: Thank you for your insightful comment regarding the biological relevance of NdTryp expression patterns and RNAi phenotypes. We agree that these important findings warrant deeper discussion, and we have substantially revised the Discussion section accordingly and removed extraneous statements.

1) Regarding the high expression of NdTryp in hepatopancreas:

We have expanded our discussion to address the biological significance of the exceptionally high NdTryp expression in hepatopancreas compared to other tissues (Page 11, Lines 299-304).

2) Regarding the RNAi-induced hepatopancreatic necrosis:

We have added the mechanistic discussion explaining how NdTryp knockdown leads to severe tissue necrosis (Pages 11-12, Lines 314-338).

---

## [Decision Letter · Decision Letter 2]

7 Jan 2026

Dear Dr. Zhang,

Thank you for submitting your manuscript to PLOS ONE. After careful consideration, we feel that it has merit but does not fully meet PLOS ONE’s publication criteria as it currently stands. Therefore, we invite you to submit a revised version of the manuscript that addresses the points raised during the review process.

1. This manuscript not technically sound, and the data cannot support the conclusions. PLOS ONE is designed to communicate primary scientific research, and welcome submissions in any applied discipline that will contribute to the base of scientific knowledge. But this manuscript not adhere to the criteria for scientific research article that results show not sufficient to support the conclusion.

2. The whole manuscript is hard to read in current status. The language in submitted articles must be clear, correct, and unambiguous.

3. The revised manuscript needs to address each of the comments of the reviewers.

We look forward to receiving your revised manuscript.

Kind regards,

Tzong-Yueh Chen, Ph.D.

Academic Editor

PLOS One

Journal Requirements:

Reviewers' comments:

Reviewer's Responses to Questions

**Comments to the Author**

Reviewer #2: (No Response)

2. Is the manuscript technically sound, and do the data support the conclusions?

Reviewer #2: Yes

3. Has the statistical analysis been performed appropriately and rigorously?

Reviewer #2: Yes

4. Have the authors made all data underlying the findings in their manuscript fully available?

Reviewer #2: Yes

5. Is the manuscript presented in an intelligible fashion and written in standard English?

Reviewer #2: No

Reviewer #2: Reviewer’s comments

Title: Physiological and biochemical characterization of trypsin from Neocaridina denticulata sinensis and its roles in ontogenesis and immune response

Manuscript Number: PONE-D-25-30024R2

Journal: PLOS ONE

This study characterizes a trypsin gene (NdTryp) from the freshwater shrimp Neocaridina denticulata sinensis through expression profiling, developmental analysis, bacterial challenge experiments, RNAi knockdown, and biochemical characterization of a recombinant protein. However, several weaknesses limit the clarity and impact of the manuscript. The integration of developmental, immune, and biochemical roles remains largely descriptive, with mechanistic links insufficiently developed. Transitions between biological findings and proposed applications are often abrupt and speculative. In addition, despite the authors’ statement that the manuscript has undergone native English editing, numerous sentences still contain inappropriate logical connectors, imprecise wording, and awkward sentence structures that obscure the intended arguments. Overall, substantial refinement of sentence logic, structural coherence, and interpretative restraint is required to strengthen the manuscript. Therefore, I recommend a “major revision” for this manuscript in PLOS ONE. Specific comments are provided below:

Specific comments

Although the authors indicate that the manuscript has undergone native English editing, we and several native English–speaking colleagues still found some sentences difficult to follow due to issues in wording, sentence structure, or logical flow. The examples noted above are intended to be illustrative rather than exhaustive. In a few instances, these language issues may also reflect minor ambiguities in the underlying academic interpretation. We therefore suggest an additional round of careful revision focusing on both linguistic clarity and conceptual precision, to ensure that the study’s arguments are communicated clearly to readers.

Abstract:

The following comments highlight selected sentences with evident issues in wording, sentence structure, or logical linkage, as illustrative examples rather than an exhaustive list of all language-related concerns in this section.

1. Line 18: “Among crustaceans, trypsin is a canonical serine protease.”

2. Line 18-20: “However, in the freshwater shrimp Neocaridina denticulata sinensis, how its contributions to ontogeny, antibacterial defense, and catalytic performance align within a single framework has not been well pinned down.”

3. Line 21-22: “Its expression was highest in the hepatopancreas, with cellular localization to storage cells (R-cells) and tubule-lining epithelial cells (ECTs).”

4. Line 22: “The signal is anatomically constrained.”

5. Line 29-30: “Taken together, these data position NdTryp as a multifunctional protease at the intersection of late embryonic development and innate antibacterial defense.”

Introduction:

The following comments highlight selected sentences with evident issues in wording, sentence structure, or logical linkage, as illustrative examples rather than an exhaustive list of all language-related concerns in this section.

1. Line 37-38: “Proteases represent one of the most abundant and diverse enzymes, responsible not only for protein degradation but also for regulating specific functional proteins through specificity to maintain the normal functioning of organisms.”

2. Lines 41–42: “Trypsin causes tissue damage due to protein hydrolysis, so trypsin is produced as an inactive precursor called trypsinogen.”

3. Lines 54–56: “A process mediated primarily by trypsin-catalyzed degradation of muscle tissue, which severely constrains shrimp shelf life during refrigerated transport.”

Materials and methods: The logical flow is acceptable; only English language editing is required.

Result: The logical flow is acceptable; only English language editing is required.

Discussion: Overall, the Discussion would be strengthened by clearer separation between data-supported conclusions and hypothesis-driven speculation, as well as by a more restrained interpretation aligned with the study’s experimental scope.

**Do you want your identity to be public for this peer review?** For information about this choice, including consent withdrawal, please see our Privacy Policy

Reviewer #2: No

---

## [Author Response · Author response to Decision Letter 3]

14 Jan 2026

Detailed Response to Reviewers

Dear Editor,

Thank you for your useful comments and suggestions on our manuscript (PONE-D-25-30024R2). We have tried our best to revise the manuscript according to the reviewers’ suggestions, and detailed corrections are listed below point by point:

Reviewer #2: Reviewer’s comments

Title: Physiological and biochemical characterization of trypsin from Neocaridina denticulata sinensis and its roles in ontogenesis and immune response

Manuscript Number: PONE-D-25-30024R2

Journal: PLOS ONE

This study characterizes a trypsin gene (NdTryp) from the freshwater shrimp Neocaridina denticulata sinensis through expression profiling, developmental analysis, bacterial challenge experiments, RNAi knockdown, and biochemical characterization of a recombinant protein. However, several weaknesses limit the clarity and impact of the manuscript. The integration of developmental, immune, and biochemical roles remains largely descriptive, with mechanistic links insufficiently developed. Transitions between biological findings and proposed applications are often abrupt and speculative. In addition, despite the authors’ statement that the manuscript has undergone native English editing, numerous sentences still contain inappropriate logical connectors, imprecise wording, and awkward sentence structures that obscure the intended arguments. Overall, substantial refinement of sentence logic, structural coherence, and interpretative restraint is required to strengthen the manuscript. Therefore, I recommend a “major revision” for this manuscript in PLOS ONE. Specific comments are provided below:

Specific comments

Although the authors indicate that the manuscript has undergone native English editing, we and several native English–speaking colleagues still found some sentences difficult to follow due to issues in wording, sentence structure, or logical flow. The examples noted above are intended to be illustrative rather than exhaustive. In a few instances, these language issues may also reflect minor ambiguities in the underlying academic interpretation. We therefore suggest an additional round of careful revision focusing on both linguistic clarity and conceptual precision, to ensure that the study’s arguments are communicated clearly to readers.

Abstract:

The following comments highlight selected sentences with evident issues in wording, sentence structure, or logical linkage, as illustrative examples rather than an exhaustive list of all language-related concerns in this section.

1. Line 18: “Among crustaceans, trypsin is a canonical serine protease.”

2. Line 18-20: “However, in the freshwater shrimp Neocaridina denticulata sinensis, how its contributions to ontogeny, antibacterial defense, and catalytic performance align within a single framework has not been well pinned down.”

3. Line 21-22: “Its expression was highest in the hepatopancreas, with cellular localization to storage cells (R-cells) and tubule-lining epithelial cells (ECTs).”

4. Line 22: “The signal is anatomically constrained.”

5. Line 29-30: “Taken together, these data position NdTryp as a multifunctional protease at the intersection of late embryonic development and innate antibacterial defense.”

Introduction:

The following comments highlight selected sentences with evident issues in wording, sentence structure, or logical linkage, as illustrative examples rather than an exhaustive list of all language-related concerns in this section.

1. Line 37-38: “Proteases represent one of the most abundant and diverse enzymes, responsible not only for protein degradation but also for regulating specific functional proteins through specificity to maintain the normal functioning of organisms.”

2. Lines 41-42: “Trypsin causes tissue damage due to protein hydrolysis, so trypsin is produced as an inactive precursor called trypsinogen.”

3. Lines 54-56: “A process mediated primarily by trypsin-catalyzed degradation of muscle tissue, which severely constrains shrimp shelf life during refrigerated transport.”

Materials and methods: The logical flow is acceptable; only English language editing is required.

Result: The logical flow is acceptable; only English language editing is required.

Discussion: Overall, the Discussion would be strengthened by clearer separation between data-supported conclusions and hypothesis-driven speculation, as well as by a more restrained interpretation aligned with the study’s experimental scope.

Response to Reviewer #2:

We sincerely thank the reviewer for the thorough evaluation and constructive suggestions. We have carefully addressed all comments and made substantial revisions to improve the manuscript quality.

For Responding to English Language Quality:

We acknowledge the reviewer’s concern about language clarity. In response:

Native English editing: We have engaged a native English-speaking colleague to comprehensively revise the manuscript, focusing on logical flow, sentence structure, and appropriate use of connectors.

Specific revisions: We have carefully revised all sentences highlighted by the reviewer in the Abstract and Introduction sections, improving: 1) Logical connectors and sentence transitions, 2) Wording precision and clarity, 3) Sentence structure and coherence. 4) In addition, we also proofread the full text of the manuscript.

For Responding to Discussion Section:

We sincerely appreciate this insightful comment. We fully agree that distinguishing between direct experimental evidence and theoretical implications is crucial for the scientific rigor of the manuscript. In the revised Discussion section, we have made the following adjustments:

Clearer Separation: We have restructured the text to explicitly distinguish conclusions directly derived from our results from broader hypotheses regarding their potential roles.

Restrained Interpretation: We have carefully refined the wording to avoid over-interpretation. Speculative statements have been tempered to ensure they strictly align with the experimental scope of this study.

Supporting References: We have added new relevant references to ground our hypothesis-driven speculations in established literature, ensuring that the discussion is both logical and well-supported.

Linguistic Clarity: Through the assistance of a native British colleague, we have optimized the logical connectors to clearly signal transitions between data analysis and scientific conjecture.

---

## [Decision Letter · Decision Letter 3]

28 Jan 2026

Physiological and biochemical characterization of trypsin from Neocaridina denticulata sinensis and its roles in ontogenesis and immune response

PONE-D-25-30024R3

Dear Dr. Zhang,

We’re pleased to inform you that your manuscript has been judged scientifically suitable for publication and will be formally accepted for publication once it meets all outstanding technical requirements.

Kind regards,

Tzong-Yueh Chen, Ph.D.

Academic Editor

PLOS One

Additional Editor Comments (optional):

Reviewers' comments:

Reviewer's Responses to Questions

**Comments to the Author**

Reviewer #2: (No Response)

2. Is the manuscript technically sound, and do the data support the conclusions?

Reviewer #2: Yes

3. Has the statistical analysis been performed appropriately and rigorously?

Reviewer #2: Yes

4. Have the authors made all data underlying the findings in their manuscript fully available?

Reviewer #2: Yes

5. Is the manuscript presented in an intelligible fashion and written in standard English?

Reviewer #2: Yes

Reviewer #2: Reviewer’s comments

Title: Physiological and biochemical characterization of trypsin from Neocaridina denticulata sinensis and its roles in ontogenesis and immune response

Manuscript Number: PONE-D-25-30024R3

Journal: PLOS One

The research work entitled “Physiological and biochemical characterization of trypsin from Neocaridina denticulata sinensis and its roles in ontogenesis and immune response” investigates the biological functions and catalytic properties of a trypsin gene (NdTryp) in the freshwater shrimp N. denticulata sinensis. The authors comprehensively characterize the spatial and developmental expression patterns of NdTryp, its inducible response to Vibrio parahaemolyticus challenge, and the physiological consequences of RNA interference–mediated knockdown. In addition, the biochemical properties of recombinant NdTryp are systematically examined, revealing broad stability across temperature and pH ranges, as well as differential modulation by metal ions. Together, the study provides integrated physiological and biochemical evidence supporting the multifunctional roles of NdTryp in late embryonic development and innate immune defense, while also highlighting its potential application in aquafeed supplementation and enzymatic biotransformation. After three rounds of revision, the manuscript is now close to an acceptable standard, with only the following two points requiring further attention.

Specific comments

1. The phylogenetic tree lacks a scale bar indicating branch length (e.g., substitutions per site), which is necessary for interpreting evolutionary distances. Please adding this information.

2. For clarity and readability, the scale bar in Fig. 7 could be displayed at a larger scale.

**Do you want your identity to be public for this peer review?** For information about this choice, including consent withdrawal, please see our Privacy Policy

Reviewer #2: No

---

## [Editor Report · Acceptance letter]

PONE-D-25-30024R3

PLOS One

Dear Dr. Zhang,

I'm pleased to inform you that your manuscript has been deemed suitable for publication in PLOS One. Congratulations! Your manuscript is now being handed over to our production team.

Kind regards,

on behalf of

Prof. Tzong-Yueh Chen

Academic Editor

PLOS One